# Kinematics and morphological correlates of descent strategies in arboreal mammals suggest early upright postures in euprimates

Severine LD Toussaint[1,2]*, Dionisios Youlatos[3], John A Nyakatura[2]

[1]Center for Research on Paleontology-Paris, Paris, France; [2]Laboratory of Comparative Zoology, Institute of Biology, Humboldt University of Berlin, Berlin, Germany; [3]Department of Zoology, School of Biology, Aristotle University of Thessaloniki, Thessaloniki, Greece

## eLife Assessment

This **valuable** study examines how mammals descend effectively and securely along vertical substrates. The conclusions from comparative analyses based on behavioral data and morphological measurements collected from 21 species across a wide range of taxa are **convincing**, making the work of interest to all biologists studying animal locomotion.

*For correspondence:
severine.toussaint@mnhn.fr

Competing interest: The authors declare that no competing interests exist.

## Abstract

Ascending and descending sloping and vertical branches are critical for arboreal locomotion and likely played a major role in early primate evolution. While most studies have focused on ascent, descending behaviors also provide insight into the functional significance of arboreal adaptations. To test how descending vertical supports of varying diameters affect locomotor abilities, we quantified postural and kinematic features during descents and ascents on vertical supports in 21 eutherian and metatherian mammals and examined their relation to morphology. Primates showed greater variability in descent behaviors, using tail-first and side postures more often than other mammals, which predominantly descended head-first. Overall, animals adopted several kinematic adjustments to enhance stability during descent compared to ascent, including slower speeds, higher duty factors, and greater use of asymmetrical gaits. Additionally, vertical descent strategies reflected trade-offs among body mass, limb proportions, and head mass. Using a morphology-based model, we then inferred possible descent behaviors in 13 extinct euarchontoglires. Our results suggest that ancestral adaptations for vertical locomotion may have promoted frequent upright (head-up) postures in early primates.

## Introduction

Arboreal environments impose strong physical constraints on vertebrate morphology and locomotor adaptations, due to the diversity of supports varying in orientation, diameter, and compliance. For arboreal mammals, the ability to navigate sloping and vertical trunks, branches, and lianas is essential, as these constitute the most available supports in trees. Climbing on vertical supports is a key aspect of arboreal locomotion in both fully arboreal and scansorial species, enabling access to the canopy from the ground, diverse food sources, enhanced vigilance, predator avoidance, and secure resting and nesting sites (*Hildebrand, 1995*). Climbing abilities in mammals date back to the Jurassic (*Zheng et al., 2013*; *Bi et al., 2014*; *Meng et al., 2015*) and have played a central role in the evolution of

**eLife digest** For many mammals living in forests, moving in trees is a daily necessity for survival. Arboreal environments offer habitats of varying sizes, orientations and flexibility, requiring animals to adopt specialized ways to move.

One particularly challenging behavior is climbing both up and down vertical supports such as tree trunks or lianas. Vertical climbing has played a significant role in mammalian evolution, especially in primates, whose ancestors developed grasping hands and feet suited for life in trees. While upward climbing has been extensively studied, descending vertically is equally crucial and potentially riskier, yet remains poorly understood.

Animals descending supports must manage their balance, speed, and body posture to prevent falling. These strategies may depend on measurable factors such as body mass, limb proportions, and grasping structures, which influence the distribution of forces along the body. Understanding how living mammals descend vertically can offer a functional framework for interpreting fossil anatomy and reconstructing the behavior of early primates and their relatives.

Toussaint, Youlatos and Nyakatura investigated whether arboreal mammals of small body mass employ consistent strategies when descending supports, and how these relate to body size and limb proportions. The researchers also examined whether these relationships make it possible to infer climbing abilities in fossils of early primates and their kin. Addressing these questions is vital because vertical movement is a high-risk yet common behavior in arboreal mammals that has shaped their ecological adaptations and morphology.

The results showed that vertical descent is not determined solely by body size but reflects distinct evolutionary strategies among arboreal mammals. Using high-speed video analysis of 1,390 descents and 1,400 ascents across 57 individuals from 21 species (including primates, rodents, carnivorans, marsupials and tree shrews), Toussaint et al. identified three descent postures: head-first, sideways and tail-first.

Non-primate mammals predominantly descended head-first, whereas primates displayed unique sideways and tail-first strategies, with variations in speed and performance. These strategies were linked to morphology, including limb proportions, tail length, and relative head size. Comparing contemporary species with fossils allowed them to reconstruct vertical descent strategies in 13 early primates and their relatives, revealing a major evolutionary shift towards upright descent postures (keeping their heads upward) throughout their history.

The findings of Toussaint et al. can benefit primatologists, evolutionary biologists and paleontologists by enhancing our understanding of how animals adapt to their environments and how fossil species can be interpreted through modern analogues. Beyond academic research, this knowledge can improve animal welfare in captivity by promoting enclosures with diverse arboreal supports of varying sizes and orientations for animals to utilize. Finally, these insights could inform the design of bio-inspired robots by identifying stable strategies for moving on vertical structures, relevant for search-and-rescue or inspection missions.

euarchontoglires, especially primates (*Wood Jones, 1916*; *Cartmill, 1985*; *Hunt et al., 1996*; *Hirasaki et al., 2000*; *Isler, 2005*; *Hanna et al., 2008*; *Hanna et al., 2017*; *Karantanis et al., 2018*; *Nyakatura, 2019*). Adaptations for climbing vertical supports have been identified in early Paleogene primates through postcranial fossil analyses (*Dagosto, 2007*; *Gebo, 2011*; *Boyer et al., 2017*; *Yapuncich et al., 2019*). Frequent vertical climbing may have been a prerequisite for the development of hallucal and pollical grasping in early primates (*Szalay and Dagosto, 1988*; *Toussaint et al., 2020*), and has been proposed as an alternative scenario to the hypothesis of cautious displacements on narrow branches of angiosperm tree peripheries (*Bloch and Boyer, 2002*; *Cartmill, 1992*; *Sussman et al., 2013*).

Several extant mammal species have been proposed as models for early primate arboreal adaptations, including small strepsirrhines (*Toussaint et al., 2015*; *Shapiro et al., 2016*), platyrrhines (*Nyakatura, 2019*; *Toussaint et al., 2020*; *Kirk et al., 2008*), treeshrews (*Sargis, 2007*), rodents (*Urbani and Youlatos, 2013*; *Orkin and Pontzer, 2011*; *Byron et al., 2011*), carnivorans (*McClearn, 1992*), and marsupials (*Toussaint et al., 2020*; *Youlatos, 2008*; *Lemelin and Schmitt, 2007*; *Shapiro and Young,*

2010; *Shapiro et al., 2014*; *Youlatos et al., 2018*). While not all arboreal mammals travel on narrow terminal branches, all rely on vertical supports to navigate canopy strata. However, few comparative studies have examined vertical climbing in non-primate arboreal mammals, and most have focused on ascents (*Hanna et al., 2017*; *Preuschoft, 2002*; *Antunes et al., 2016*; *Clemente et al., 2019*). Although behaviors, such as dropping, jumping, or gliding allow arboreal animals to move to lower tree layers, the ability to safely descend sloping and vertical supports remains critical, yet largely understudied (*Birn-Jeffery and Higham, 2014*).

Non-primate arboreal mammals often adopt a head-first posture when descending vertical supports, as reported in some rodents (*Karantanis et al., 2018*; *Youlatos, 2011*; *Youlatos and Panyutina, 2014*) and procyonid carnivorans (*McClearn, 1992*; *Jenkins and McClearn, 1984*). This strategy relies on reversed feet and functional claws that enable secure gripping of tree bark during descent. Head-first descent is also common in prehensile-tailed mammals, such as certain platyrrhine primates (*Youlatos and Gase, 1994*; *Garber and Rehg, 1999*; *Lawler and Stamps, 2002*) and the kinkajou (*McClearn, 1992*). In species lacking a prehensile tail, and in primates that lack claws and use their digits to wrap around supports, alternative descent strategies are required. In a study of nine small to medium-sized strepsirrhines, *Perchalski, 2021* found that lemurs under 1 kg predominantly used head-first descents regardless of support properties, whereas larger species (>1 kg) increasingly adopted tail-first postures on supports angled at 45° or more, depending on species. Both head-first and tail-first strategies involve orthograde postures with the body aligned parallel to the support (*Hunt et al., 1996*). Perchalski also recorded other descent behaviors, including leaping, dropping, and asymmetrical descents that did not fit into strict head-first or tail-first categories. These findings suggest that in strepsirrhines, body mass and locomotor adaptations influence the choice of descent posture. However, such behavioral variation also highlights the need for broader comparative data across arboreal mammals to fully understand the functional and ecological significance of vertical descent strategies in an evolutionary context.

From a locomotor perspective, speed and duty factor (DF, the percentage of stride duration a limb remains in contact with the support) are key indicators of arboreal locomotor performance (*Cartmill et al., 2007a*). Studies across tetrapods have shown that animals generally move more slowly on narrow supports than on wider ones (*Lammers and Stakes, 2025*; *Schmitt and Lemelin, 2002*). Small primates (*Shapiro et al., 2016*; *Hesse et al., 2015*; *Nyakatura et al., 2008*), rodents (*Karantanis et al., 2018*; *Karantanis et al., 2017*; *Wölfer et al., 2021*), and marsupials (*Shapiro et al., 2014*; *Lammers et al., 2006*) typically decrease speed and increase duty factor, especially for the forelimbs, during descents compared to ascents on inclined supports and compared to locomotion on horizontal supports. These mechanisms supposedly help enhance stability by allowing more cautious displacements (*Birn-Jeffery and Higham, 2014*; *Lammers and Stakes, 2025*; *Wölfer et al., 2021*).

Gait patterns also reflect arboreal adaptations, particularly in primates (*Preuschoft, 2002*; *Cartmill et al., 2007a*; *Vilensky and Larson, 1989*; *Hildebrand, 1967*). Vertical climbing employs both symmetrical (walk, trot, run) and asymmetrical (bound, half-bound, gallop) gaits, similar to above-branch locomotion. Symmetrical gaits are defined as locomotor patterns in which the footfalls of a girdle (a pair of fore- or hindlimbs) are evenly spaced in time, with the right and left limbs of a pair of limbs being approximately 50% out of phase with each other (*Hildebrand, 1967*; *Hildebrand, 1966*). Symmetrical gaits can be further divided into two types: diagonal-sequence gaits, in which a hindlimb footfall is followed by that of the contralateral forelimb, and lateral-sequence gaits, in which a hindlimb footfall is followed by that of the ipsilateral forelimb (*Hildebrand, 1967*; *Shapiro and Raichlen, 2005*; *Cartmill et al., 2007b*). In contrast, asymmetrical gaits are characterized by unevenly spaced footfalls within a girdle, with the right and left limbs moving in near synchrony (*Hildebrand, 1977*). Asymmetrical gaits, often used by small mammals at high speeds (*Lammers and Stakes, 2025*; *Dunham et al., 2020*), allow for an increase in the distance traveled but often include an aerial phase (*Hildebrand, 1977*), raising support reaction forces and potentially causing oscillations or breakage on narrow horizontal or sloping branches, complicating locomotor control and safety. Benefits of asymmetrical gaits have thus been proposed to be related to both body mass and support diameter (*Chadwell and Young, 2015*). On vertical supports, such as trunks or lianas, these risks are reduced due to different force distributions and gravity orientation. Symmetrical gaits are widespread in primates and have been linked to the use of the fine branch niche, as enhanced grasping and hindlimb dominance supposedly increase stability during cautious locomotion on narrow horizontal

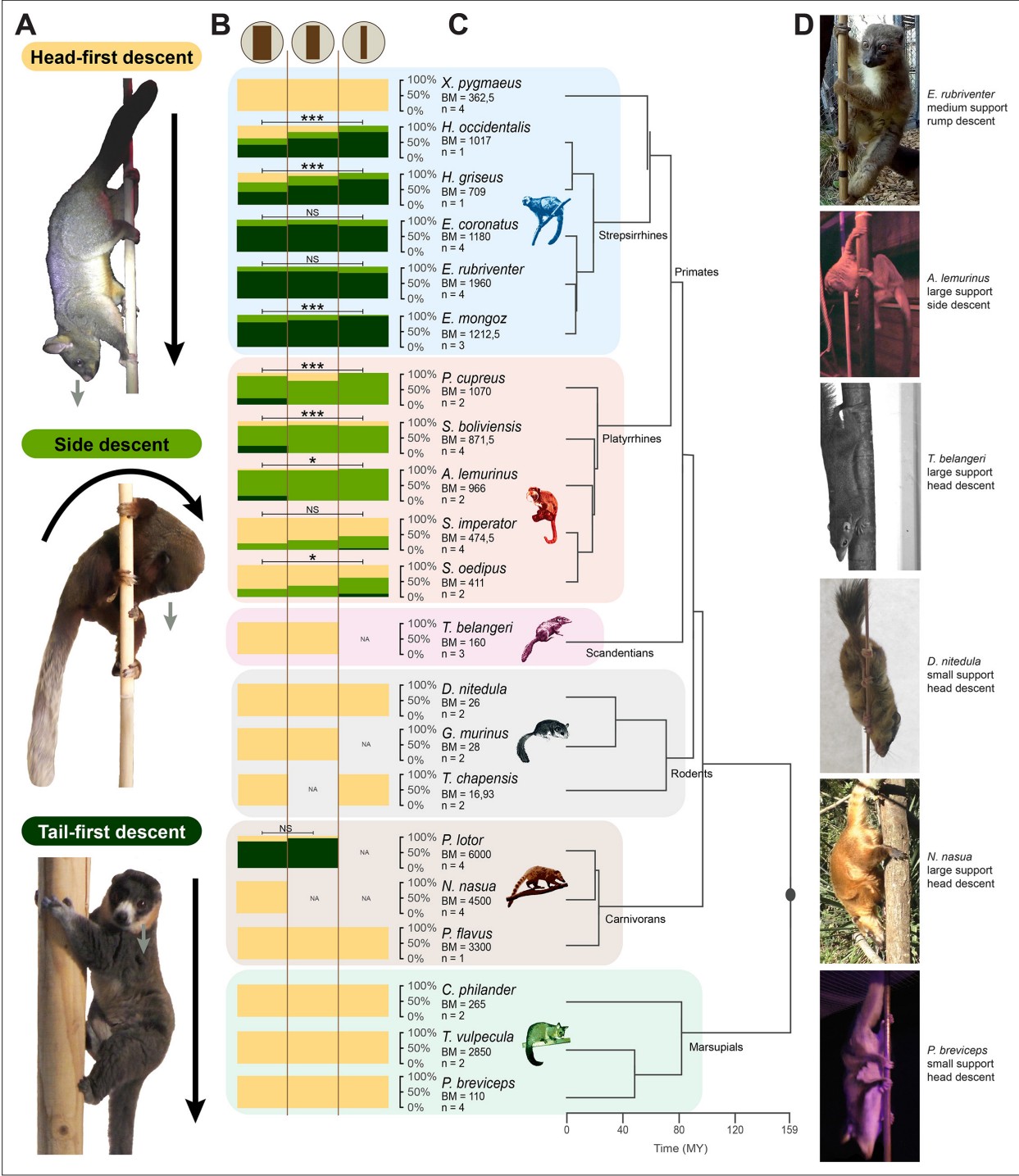

**Figure 1.** Strategies of descent on vertical supports of various diameters by species. (**A**) Photographs illustrating the three different strategies of descent identified in this study. Black arrows represent the axis of the body regarding the support and the direction of the movement, and gray arrows represent the direction of the gaze. (**B**) Proportions (in %) of each descent type (yellow = head-first descent, light green = side descent, and dark green = tail-first descent) occurring on each support category (vertical branches of large, medium, and small diameters represented by brown icons at the top of the graph), by species. Proportions of all individuals of the same species (n) were averaged. See *Supplementary file 1* for details on the individuals studied, and *Supplementary file 2* for the mean body masses (BM) references. NA = no data for the given condition. Results of statistical tests between support diameters (ANOVAs or MANOVAs on bootstrapped samples, confirmed by post hoc tests) are represented with annotations defined as NS (non-significant): p>0.05, *p<0.05, **p<0.01, and ***p<0.001. See *Supplementary file 5A* for associated p-values. (**C**) Associated phylogeny of the species studied with branch length obtained from http://timetree.org/; *Kumar et al., 2017*. (**D**) Photographs of individuals of various species descending vertical supports of different diameters.

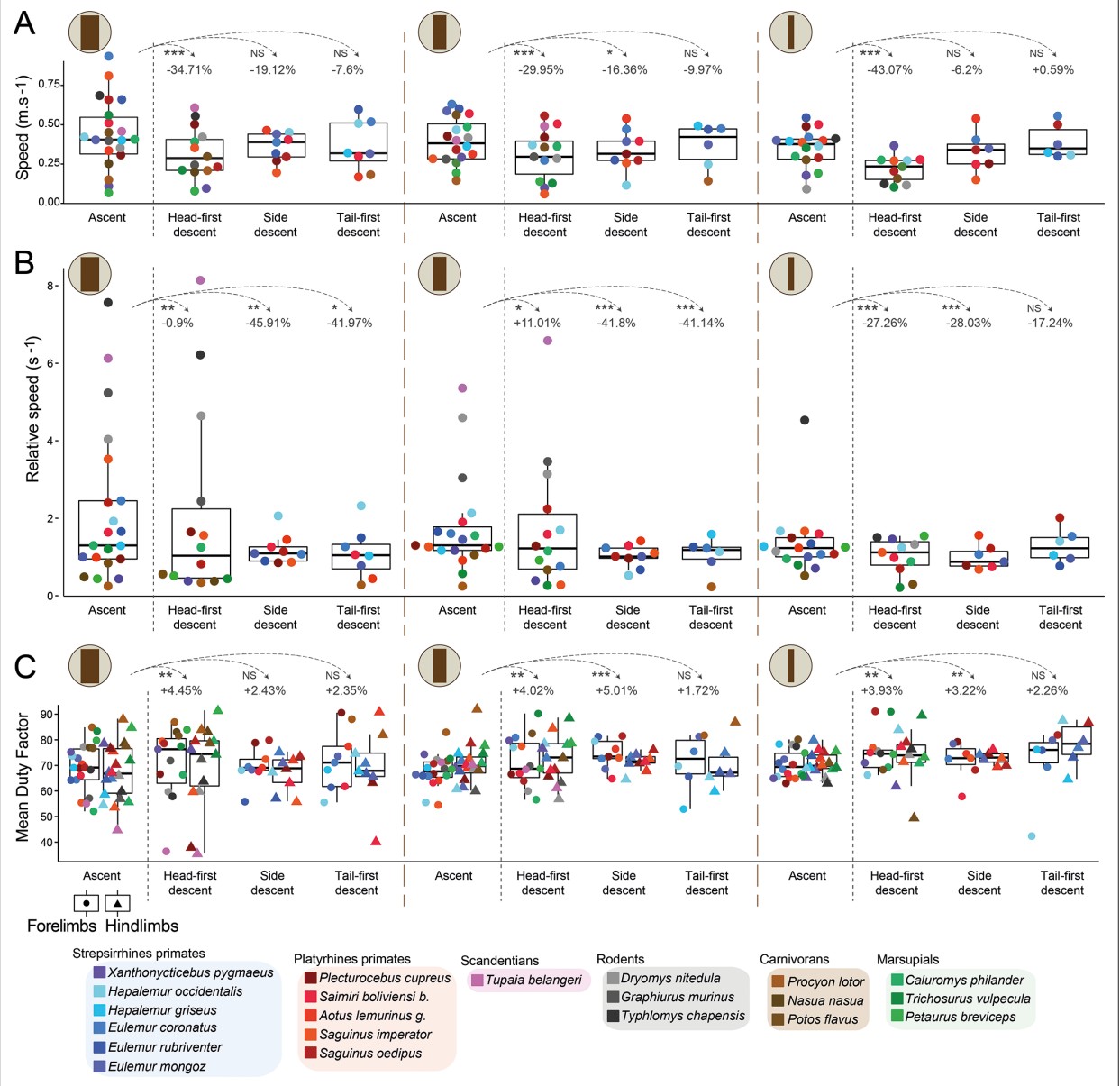

**Figure 2.** Kinematics of locomotion during ascents and descents on vertical supports of varying diameters. (**A**) Mean absolute speeds by species during ascents and each descent strategy on the vertical supports of large, medium, and small diameters (represented by brown icons on the top left of the plots), with box plot representations of the medians for all species combined in ascents and each descent strategy. (**B**) Mean relative speeds (based on individuals' body length) by species during ascents and each descent strategy on each support type, with box plot representations of the medians for all species combined in ascents and each descent strategy. (**C**) Mean duty factors by species, differentiating forelimbs (left box plots) and hindlimbs (right box plots), during ascents and each descent strategy on each support type, with box plot representations of the medians for all species combined in ascents and each descent strategy. Results of statistical tests between ascents and each descent strategy (two-sided Mann-Whitney U tests) are represented with annotations defined as NS: $p>0.05$, *$p<0.05$, **$p<0.01$, and ***$p<0.001$. See *Supplementary file 5B–D* for associated p-values. Percentages correspond to the relative variation of each kinematic variable mean in each descent condition compared to ascents.

or sloping supports (*Hildebrand, 1967*; *Cartmill et al., 2007b*; *Cartmill et al., 2002*; *Lemelin and Cartmill, 2010*). Notably, primates adjust their symmetrical gaits with support inclination, favoring diagonal-sequence gaits during ascents and lateral-sequence gaits during descents (*Nyakatura et al., 2008*; *Granatosky et al., 2019*; *Nyakatura and Heymann, 2010*; *Prost and Sussman, 1969*; *Rollinson and Martin, 1981*; *Shapiro et al., 2025*). Symmetrical gaits are also common in rodents (*Karantanis et al., 2018*; *Karantanis et al., 2017*), scandentians (*Granatosky et al., 2022*), carnivorans (*McClearn, 1992*), and marsupials (*Cartmill et al., 2020*; *Lemelin et al., 2003*; *Gaschk et al., 2019*;

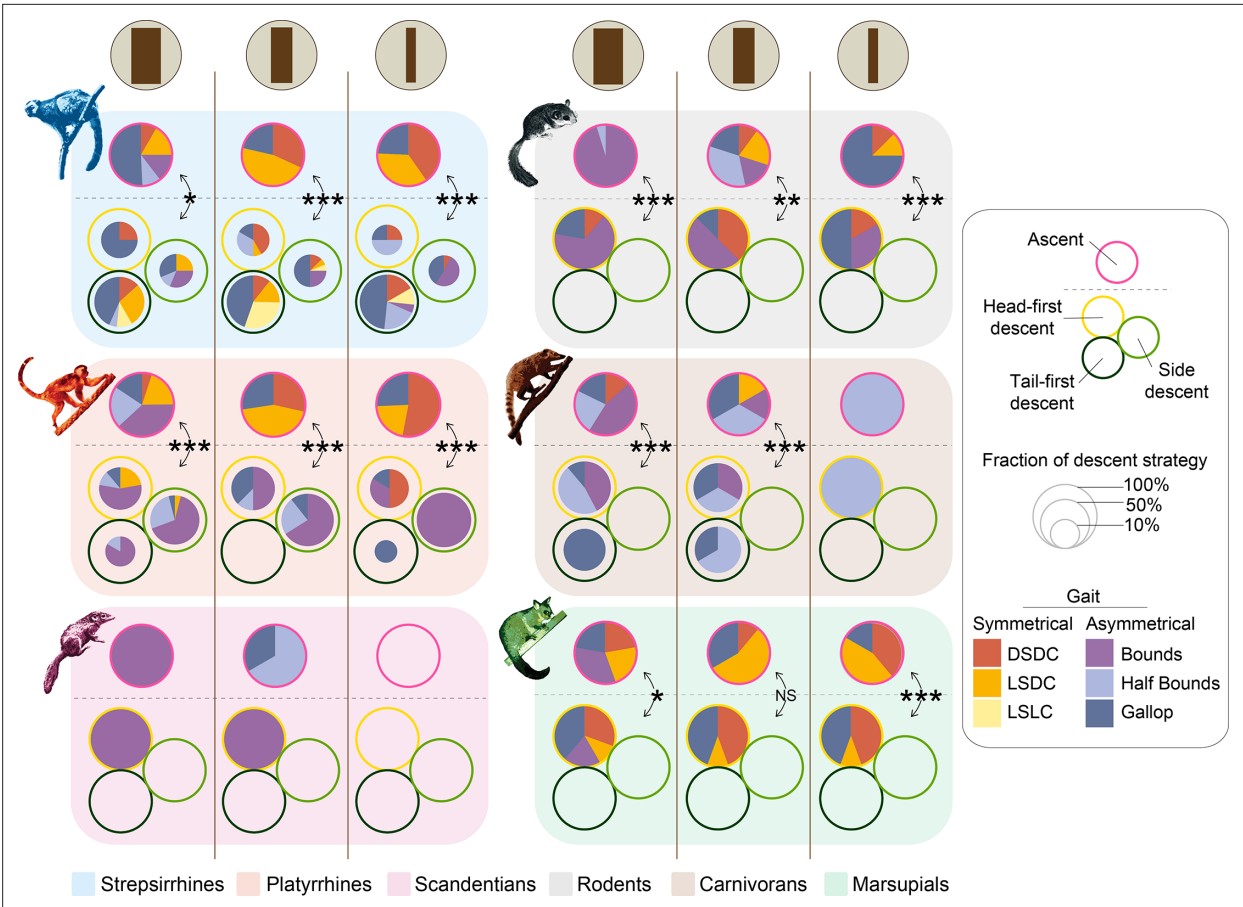

**Figure 3.** Proportions (in %) of gait types by support diameters for ascents and each descent strategy, for all phylogenetic groups. Results of statistical tests on the fraction of symmetrical versus asymmetrical gaits by group between ascents and descents on each support (two-sided Wilcoxon signed rank tests) are represented with annotations defined as NS: $p > 0.05$, *$p < 0.05$, **$p < 0.01$, and ***$p < 0.001$. NA = not applicable. DSDC: Diagonal sequence - diagonal couplet, LSDC: Lateral sequence – diagonal couplet, LSLC: Lateral sequence – lateral couplet. See **Supplementary file 5F** for associated post hoc corrected p-values. See **Figure 3—figure supplement 1** for detailed proportions of gait types by species and support diameters.

The online version of this article includes the following figure supplement(s) for figure 3:

**Figure supplement 1.** Mean proportions of gait types (in %) by support diameter and by ascent and descent conditions, for each species.

---

Karantanis et al., 2015). However, most kinematic studies have focused on horizontal or moderately sloped supports (~45°), rarely addressing vertical (~90°) supports or analyzing both symmetrical and asymmetrical gaits comprehensively. As a result, comparative data on how vertical supports of varying diameters affect descent and ascent kinematics remain limited, hindering our understanding of this common arboreal behavior.

In addition to body mass, limb proportions influence locomotor and postural strategies (**Perchalski, 2021**; **Granatosky et al., 2019**). The relative length of autopodials, particularly manual and pedal digits, is closely linked to grasping ability and thus affects locomotor performance in primates (**Nyakatura, 2019**; **Toussaint et al., 2020**; **Cartmill, 1992**; **Kirk et al., 2008**; **Lemelin and Jungers, 2007**) and small marsupials (**Toussaint et al., 2020**; **Lemelin and Schmitt, 2007**). Species with relatively short autopods may struggle to maintain a secure grip during vertical descent. Fore- and hindlimb length ratios have also been proposed as predictors of postural adaptation in vertical locomotion, with shorter forelimbs generating greater hindlimb traction during head-first descents (**Preuschoft, 2002**; **Perchalski, 2021**). Consequently, a lower intermembral index (forelimb-to-hindlimb length ratio) increases the risk of forward pitching when descending vertically head-first compared to tail-first (**Perchalski, 2021**).

Variation in weight distribution along the body, notably caused by head mass differences, may also influence postural strategies during vertical descent (i.e. head-first vs. tail-first). Primates exhibit the

highest brain-to-body mass ratio among mammals (*Boddy et al., 2012*). This relative brain expansion, traced back to early primate evolution, is a key synapomorphy alongside a divergent hallux and pollex, nails replacing claws, elongated hindlimbs and autopods, and large convergent eyes (*Cartmill, 1992*; *Szalay, 1968*; *Martin, 1990*). However, the evolutionary sequence and interplay between locomotor and cognitive specializations remain unresolved, despite consensus on the small body size (*Nyakatura, 2019*; *Dagosto et al., 2018*) and arboreality (*Upham et al., 2019*; *Hughes et al., 2021*; *Dunn et al., 2016*) of primate ancestors. While brain enlargement likely increases relative head mass, its effects on arboreal locomotion have yet to be investigated in primates or other mammals. Given that head mass influences both the center of mass and moment of inertia, greater head mass may increase toppling risk during vertical head-first descent.

In this study, we aim to test (a) whether postural and kinematic adjustments during vertical descents versus ascents vary consistently across mammalian groups; (b) whether these strategies correlate with specific morphological parameters; and (c) whether such correlations can be used to infer descent postures in fossils, shedding light on the evolutionary history of vertical arboreal locomotion, particularly in primates. To address these questions, we analyzed locomotor sequences of descents and ascents on vertical supports in 21 small- to medium-sized arboreal mammal species, including taxa proposed as modern analogues of early primates. We first examined the effect of support diameter (small, medium, large) on postural strategies and kinematics (speed, duty factor, gait) during descents relative to ascents. We then assessed the relationships between descent strategies and morphological parameters (body mass, limb proportions, and relative head mass) for each species. These data allowed us to build a model incorporating seven significant morphological predictors, which we applied to infer vertical descent behavior in 13 extinct euarchontoglires, including adapiforms (early strepsirrhines), omomyiforms (early haplorhines), a sciuromorph (early rodent), and plesiadapiforms (stem primates).

We predict that:

(H1) Postural strategies: species under 1 kg and/or possessing a prehensile tail will exhibit head-first descent postures more frequently than larger species, which will favor tail-first postures (*Perchalski, 2021*).

(H2) Kinematic adjustments: animals will adopt more cautious locomotor patterns during vertical descents compared to ascents, characterized by reduced speed, increased duty factors, and a higher proportion of symmetrical gaits, particularly on narrow supports (*Lammers and Stakes, 2025*; *Karantanis et al., 2017*; *Wölfer et al., 2021*; *Granatosky et al., 2019*; *Granatosky et al., 2022*).

(H3) Morphological correlates: the intermembral index will positively correlate with the proportion of head-first descents, whereas relative autopod length and head size will correlate negatively with head-first descent frequency (*Cartmill, 1992*; *Preuschoft, 2002*; *Perchalski, 2021*).

(H4) Evolutionary implications: early euarchontoglires, due to their small size and generalized locomotor abilities, likely employed a high proportion of head-first descents (*Karantanis et al., 2018*; *Wölfer et al., 2021*), whereas early euprimates may have increasingly favored tail-first descents associated with larger body and brain, and elongated hindlimbs and autopods (*Cartmill, 1992*; *Szalay, 1968*; *Martin, 1990*).

## Results

### Descent strategies on vertical supports among arboreal mammals

We recorded the locomotion of 57 adult individuals from 21 species: 6 strepsirrhine primates, 5 platyrrhine primates, 1 scandentian, 3 rodents, 3 carnivorans, and 3 marsupials (*Figure 1*, *Supplementary file 1*, *Source data 1*). We categorized supports in three diameters: small (i.e. twigs), medium (i.e. branches), and large (i.e. boughs and very large trunks) relative to the size of the animals' hands and feet (see section Materials and methods for more details). Whenever possible, we analyzed up to 10 ascents and 10 descents per individual on each support type. In total, we analyzed 1400 ascents and 1390 descents.

We identified three distinct strategies of descent: head-first descent, side descent, and tail-first descent (*Figure 1A*), which partly overlap with Perchalski's ethogram (*Perchalski, 2021*). During a head-first descent, the body is positioned head-down, parallel to the support, hands and feet are anchored to the support, and the face is directed downward (see *Trichosurus vulpecula* in *Figure 1A*).

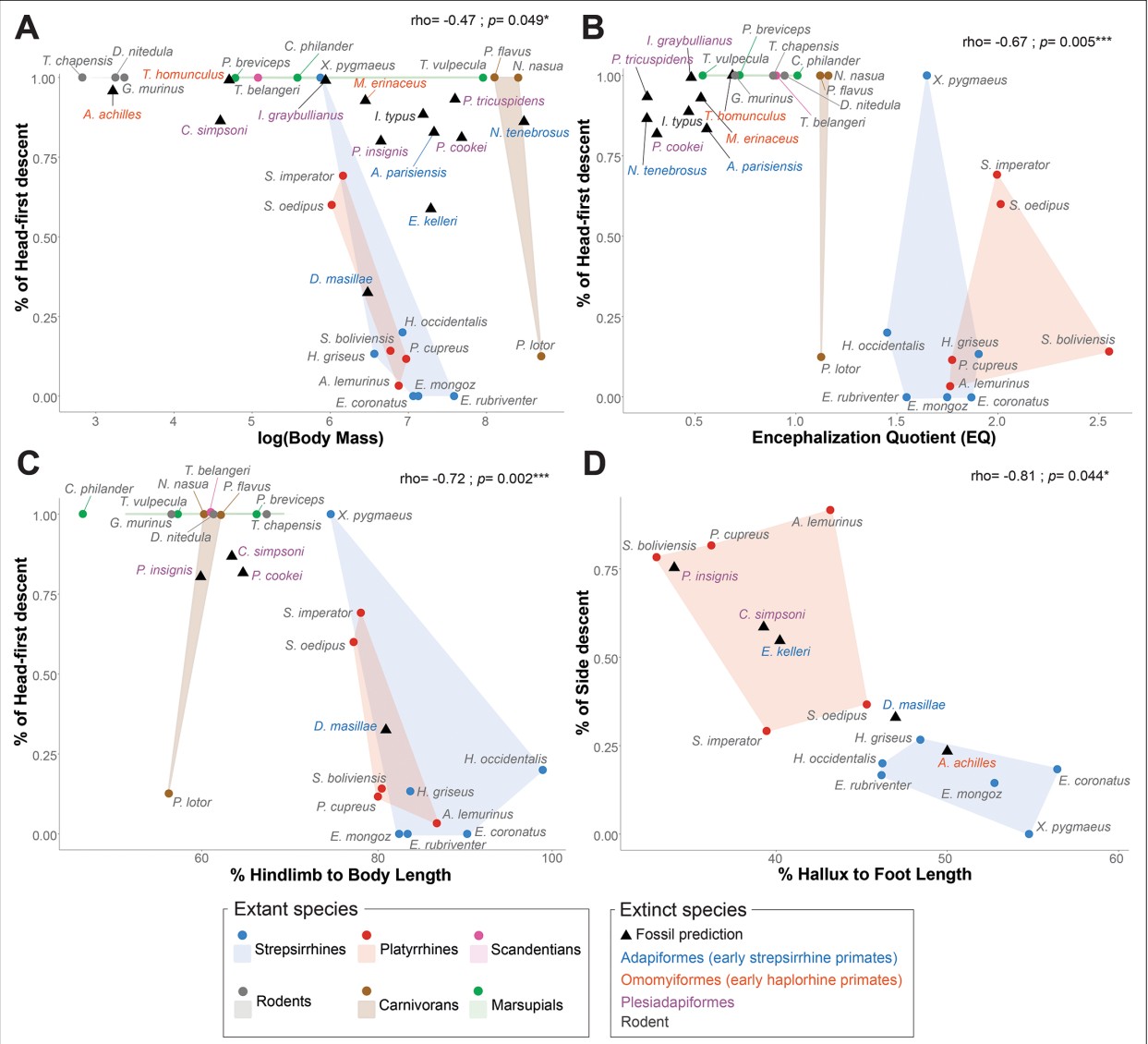

**Figure 4.** Morphological correlates with vertical descents and prediction of descent strategy in extinct species. Spearman correlations and associated corrected p-values between the proportion of head descents by extant species and (**A**) the logarithm of mean body mass, (**B**) the mean encephalization quotient (EQ), (**C**) the hindlimb to body length ratio (in %), and (**D**) between the proportion of side descent in extant primate species only and the hallux to foot length ratio (in %). Each plot also includes the head descent proportion (**A–C**) and side descent proportion (**D**) of fossils predicted with the Schafer's multiple imputation procedure. See (**Table 1**) for body measurements definitions and (**Supplementary files 2 and 3**) for mean body masses, endocranial volume (ECV), and calculated EQ by species. See (**Figure 4—figure supplement 2**) for the plots of the percentage of head descent against the six other morphological variables analyzed by species, and (**Supplementary file 5H and I**) for Spearman rhos and associated corrected p-values between each morphological variable and the proportion of, respectively, head (**Supplementary file 5H**) and side descents (**Supplementary file 5I**) on vertical supports. See **Supplementary file 4** for calculated morphological proportions of extinct species and **Supplementary file 5J** for the resulting predictions of their descent behavior.

The online version of this article includes the following figure supplement(s) for figure 4:

**Figure supplement 1.** Major axis orthogonal regressions on the log-transformed mean brain and mean body masses for all species (in black with the gray area representing the associated confidence interval) and by phylogenetic groups color coded as defined in **Figure 1C**.

**Figure supplement 2.** Spearman correlations and associated corrected p-values between the proportion of head-first descents by species on vertical supports (all diameters combined) and the relative: (**A**) forelimb length, (**B**) intermembral index, (**C**) tail length, (**D**) thumb length, (**E**) hand length, and (**F**) foot length.

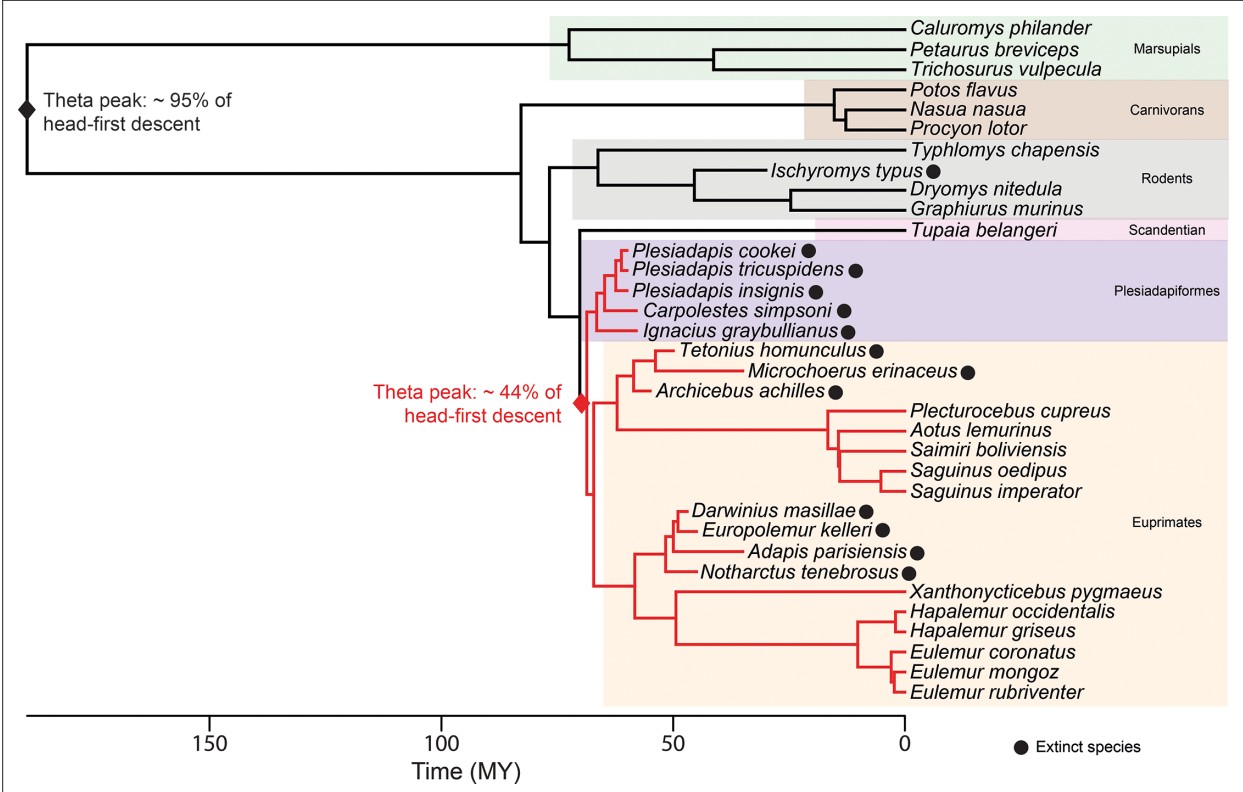

**Figure 5.** Phylogeny of studied species, including extant and extinct taxa, and representation of the best evolutionary model of head-first descent proportion. We calculated a consensus tree for extant species based on 1000 trees from https://vertlife.org/, to which we added extinct taxa using age and phylogenetic topology from literature. See (*Supplementary file 2*) for estimated age of extinct species and associated references. The best evolutionary model of head-first descent proportion is an Ornstein-Uhlenbeck evolutionary model with two optimums (OUM) with a peak shift occurring at the node 'plesiadapiformes + euprimates.' The associated theta peaks of head-first descent proportions of the two subgroups 'plesidapiforms + euprimates' vs 'others' are represented in red and black, respectively.

During a side descent, the body is rotated perpendicularly to the support and the face is also directed downward (see *Plecturocebus cupreus* in *Figure 1A*). During a tail-first descent, the body is positioned head-up and parallel to the support similarly to ascents, and the face is either directed horizontally or downward (see *Eulemur mongoz* in *Figure 1A*). We found a clear difference between primates and non-primates in the occurrence of each descent strategy on each support type (*Figure 1B*). Most non-primate species, although very diverse in body mass and morphology, showed only head-first descents independently of the orientation and diameter of the supports. The raccoons, the largest species of this study (~6 kg), were an exception and mainly exhibited tail-first descents. The other studied carnivorans, i.e., the coatis and the kinkajous (the latter being the only prehensile-tailed of this study along with the brushtail possum *Trichosurus vulpecula*), as well as all marsupials, rodents, and treeshrews exhibited only head-first descents independently of the support diameter (*Figure 1B, D*). In sharp contrast, primates rarely used head-first descents on vertical supports except for lorises (*Xanthonyc-ticebus pygmaeus,* strepsirrhine) and tamarins (*Saguinus imperator* and *S. oedipus,* platyrrhines), the smallest primate species of this study (~360 g to ~470 g, respectively). The other strepsirrhines (i.e., lemurs) descended vertical supports mainly using tail-first descents with higher proportions on small supports, and the other platyrrhines generally used side descents with also increased proportions on small supports. The difference between phylogenetic groups is substantiated by a significant phylogenetic signal contained in the overall proportion of head-first descent strategy on vertical supports ($k=0.58$, $p=6.6.10^{-4}$, effect size = 2.57).

## Kinematic adaptations during ascents and descents on vertical supports

For each complete ascent and descent sequence of locomotion, characterized by successive lift-off, touchdown, and following lift-off of all four limbs, we quantified: the absolute speed, the relative speed (in relation to the individual' body length), the forelimbs and hindlimbs' duty factors (*Figure 2*, *Source data 1*), and the gait type (categorized as symmetrical or asymmetrical, *Figure 3*, *Source data 1*).

We found a significant general tendency for all species to reduce their absolute speed by 30 to 43% when descending head-first on vertical supports compared to ascents, regardless of support diameter (*Figure 2A*, *Supplementary file 5B*). In contrast, primates did not significantly reduce their speed compared to ascents when descending sideways or tail-first (*Figure 2A*, *Supplementary file 5B*). On the contrary, in primates, the relative speeds decreased during side and tail-first descent on all supports by between 17 and 46%, indicating that animals tended to decrease the distance travelled compared to ascents (*Figure 2B*, *Supplementary file 5C*). But during head-first descents, the relative speeds tended to increase by ~10% on medium supports, while they decreased on small and large supports. It is interesting to note that the smallest species of our dataset, i.e., rodents and treeshrews (<200 g), exhibited the highest relative speeds compared to other taxa, thus travelling larger distances in proportion to their size (*Figure 2B*). On the other hand, all studied animals tended to increase both forelimbs and hindlimbs duty factors during descents compared to ascents, particularly during head-first descents on all supports by ~4%, and during side descents on medium and small vertical supports, but not when descending tail-first (*Figure 2C*, *Supplementary file 5D*).

We found that gait contains a low, yet significant phylogenetic signal ($K_{mult}$ = 0.238; $p$=4.31.10$^{-2}$; Effect Size = 1.72), suggesting the impact of shared evolutionary history of related species on locomotor characteristics. We additionally found that there is a significant similarity between phylogenetically related species and dissimilarity across phylogenetically distant groups, allowing us to merge the gait data at the phylogenetic group level (as defined in *Figure 1C*) to obtain a more general picture of the large dataset (MANOVA: $p$=2.09.10$^{-2}$, Wilks' $\lambda$ =3.36.10$^{-3}$, df = 5, *Supplementary file 5E*). Considering the difference in the proportions of descent strategies obtained between strepsirrhines and platyrrhines, we decided to keep these two subgroups separate rather than grouping all primates together. When analyzing the effect of vertical descent strategies on gait types by phylogenetic groups, we found that animals exhibited a mosaic of gait types depending on the species, support diameter, and direction of movement. Interestingly, the proportion of asymmetrical gaits significantly increased during descents on all supports for all groups studied (*Figure 3*, *Supplementary file 5F*). In effect, during vertical descents, the animals exhibited more bounds, gallop, and related asymmetrical gaits with relatively high duty factors, not only during side and tail-first descents but also during head-first descents (*Figure 3*). Primates and marsupials overall exhibited a higher proportion of symmetrical gaits compared to the other studied mammals and particularly diagonal and lateral sequence - diagonal couplets (DSDC and LSDC, respectively), especially during vertical ascents (*Figure 3*, see *Figure 3—figure supplement 1* for detailed gait types by species and support diameters). We also documented symmetrical gaits in rodents and carnivorans. Primates, marsupials, and rodents exhibited more symmetrical gaits during ascents on vertical supports of small and medium diameter compared to large vertical supports. Carnivorans, especially raccoons, exhibited a relatively small proportion of symmetrical gaits on large and medium supports during ascents, and treeshrews used only asymmetrical gaits during both ascents and descents on vertical supports of medium and large diameter. Furthermore, strepsirrhines exhibited some lateral sequence – lateral couplets (LSLC) only during vertical descents, and rodents and carnivorans exhibited more DSDC during descents and more LSDC during ascents.

## Correlations between vertical descent strategies and morphological features

To investigate links between vertical descent strategies and morphological parameters, we collected ten morphological variables, including body mass (*Supplementary file 2*), eight intrinsic body proportions calculated from measurements taken on each studied animal (*Table 1*), and relative head mass of each species, approximated by its encephalization quotient (EQ) (*Supplementary file 2*). We used the EQ (*Jerison, 1973*) as it quantifies whether the brain mass of a given species is lower (<1) or

**Table 1.** Definitions of body measurements performed on extant and extinct individuals and calculation of limb proportions.

| Length measured | Description |
| --- | --- |
| Body length | Total body length without tail (from nose to rump) |
| Tail length | Total tail length |
| Forelimb length | Total length of the forelimb (stylopod + zeugopod + autopod) |
| Hindlimb length | Total length of the hindlimb (stylopod + zeugopod + autopod) |
| Hand length | Maximal length from the tip of the third digit to the proximal carpus |
| Foot length | Maximal length from the tip of the third digit to the proximal tarsus |
| Pollex length | Maximal length from the tip of the distal phalanx to the proximal part of the metacarpus |
| Hallux length | Maximal length from the tip of the distal phalanx to the proximal part of the metatarsus |
| **Proportion calculated** | **Description** |
| % Tail to body length | $\frac{Tail\ length \times 100}{Body\ length}$ |
| % Forelimb to body length | $\frac{Forelimb\ length \times 100}{Body\ length}$ |
| % Hindlimb to body length | $\frac{Hindlimb\ length \times 100}{Body\ length}$ |
| Intermembral index | $\frac{Fore\ limb\ length\ without\ hand \times 100}{Hindlimb\ length\ without\ foot}$ |
| % Hand to forelimb length | $\frac{Hand\ length \times 100}{Forelimb\ length\ without\ hand}$ |
| % Foot to hindlimb length | $\frac{Foot\ length \times 100}{Hindlimb\ length\ without\ foot}$ |
| % Pollex to hand length | $\frac{Pollex\ length \times 100}{Hnad\ length}$ |
| % Hallux to foot length | $\frac{Hallux\ length \times 100}{Foot\ length}$ |

higher (>1) than expected from a general brain-to-body allometric relationship. We calculated the mean EQ for each species from their mean endocranial volumes, collected either from the literature or from our own measurements for previously documented species (*Supplementary files 2 and 3*). To ensure that the EQs of the different species studied are comparable and meaningful, we tested the allometry between the brain and body masses in our dataset following *Martin and Barbour, 1989* and found a significant and positive slope for all species (major axis orthogonal regression on log-transformed values: slope = 0.36, $r^2$=0.92, $p$=5.0.10$^{-12}$), indicating a negative allometry ($r$=–0.97, df = 19, $p$=2.0.10$^{-13}$), and similar allometric coefficients when restricting the analysis to phylogenetic groups (*Figure 4—figure supplement 1*).

Among the ten morphological variables analyzed, we found that the forelimb and hindlimb relative lengths correlate with each other and with the mean EQ, and the hindlimb and tail relative lengths correlate with each other (*Supplementary file 5G*). Moreover, we found that body mass, EQ, relative forelimb and hindlimb lengths, and relative tail length significantly and positively correlate with the proportion of head-first descent strategy on vertical supports, while the intermembral index significantly and negatively correlates with the proportion of head-first descent strategy (all diameters combined) (*Figure 4A–C*, *Figure 4—figure supplement 2A-C Supplementary file 5H*). Also, in primates, only the relative length of the hallux correlates significantly and negatively with the proportion of side descents (*Figure 4D*, *Supplementary file 5I*). Primates stand out from the other studied groups in having relatively longer hindlimbs (>70% of body length, *Figure 4C*), forelimbs (>70% of body length, *Figure 4—figure supplement 2A*), tails (>90% of body length, except for the lorises who have a very reduced tail, *Figure 4—figure supplement 2C*), and high EQs (>1.5, *Figure 4B*). In

comparison, carnivorans, albeit being the largest studied species (>3 kg, *Figure 4A*), possess relatively shorter limbs (~60% of body length, *Figure 5A*), and medium EQs (>1 and<1.2, *Figure 5B*). The studied rodents, scandentians, and marsupials also all possess relatively shorter limbs (<70% of body length, *Figure 4C*, *Figure 4—figure supplement 2A*), and tails (<90% of body length, *Figure 4— figure supplement 2C*), and show small to medium body masses (<3 kg), and low EQs (≤1).

## Integrating extant behavioral and morphological data to hypothesize vertical descent strategy in early euarchontoglires

Based on these results, we attempted to infer the likely descent strategies of extinct species, using their known body proportions. We selected 13 fossils relevant to the understanding of early primate evolution, including 7 euprimates, 5 plesiadapiforms, and 1 rodent with relatively complete postcranial and/or cranial remains (*Supplementary file 2*, *Supplementary file 4*). We used a Multiple Imputation (MI) method to predict descent behaviors given partial fossil morphological data. Following the benchmark of Clavel and colleagues (*Clavel et al., 2014*), which compared several MI methods for the precise problem of morphometric fossil estimation, we used Schafer's MI procedure from the R package *norm*, with 5000 runs of 50 steps Markov Chains. We calculated the prediction of head-first descent proportion on vertical supports (all diameters combined) for all fossils, using all extant and extinct species, and retaining all six morphological variables that significantly correlate with head-first descent (*Figure 4A–C*; *Figure 4—figure supplement 2A–C*). We also computed the prediction of side descents only for primate and plesiadapiform fossils, retaining all extant primate species and the respective correlated variable, i.e., the relative length of the hallux (*Figure 4D*).

Fossil species were mostly reconstructed as employing a relatively high proportion of vertical head-first descent strategy (>75%), except for *Darwinius masillae* and *Europolemur kelleri,* two medium-sized (660 g and ~1.5 kg, respectively) adapiformes (early strepsirrhine primates), that were estimated at around 30% and 60% of vertical head-first descent strategy, respectively (*Figure 4A, C*, *Figure 4— figure supplement 2A-C*; *Supplementary file 5J*). *Darwinius* remains are very well preserved, and we were able to calculate all the morphological variables for this species. For *Europolemur*, however, we could only include the estimated body mass and foot proportions in the analysis. The other adapiforms of this study, *Adapis parisiensis* and *Notharctus tenebrosus*, were reconstructed with high head-first descent proportions (~83% and~86%, respectively) despite being larger (~1.5 kg and ~9 kg, respectively). However, we could not find suitable illustrations of the known postcranial elements of these species in the literature that could be reliably incorporated into this study. Thus, we only included their reconstructed body mass and EQ, the latter being relatively low (0.56 and 0.26, respectively) compared with extant strepsirrhines. Interestingly, the omomyiforms (early haplorrhines) studied here, *Archicebus achilles*, *Microchoerus erinaceus*, and *Tetonius homunculus* were reconstructed with very high head-first vertical descent proportions (>90%, *Figure 4A, B*, *Supplementary file 5J*), but they are also known mainly from their cranial remains and only body mass and EQ were used in the head-first descent model. Yet, their EQs are also very low (<1) compared with those of extant haplorrhines (*Figure 4B*, *Supplementary file 2*, *Supplementary file 5J*). Concerning the plesiadapiformes and the fossil rodent studied, which display variable body masses ranging from the small *Carpolestes simpsoni* (~100 g) to the medium-sized plesiadapids (~2 kg, *Figure 4A*, *Supplementary file 2*), they were all reconstructed as exhibiting a relatively high proportion of head-first descent strategy (>75%, *Supplementary file 5J*). Those species possess relatively elongated limbs whose proportions fall between extant non-primates and primates (*Figure 4C*, *Figure 4—figure supplement 2A, B*, *Supplementary file 4*), but retain a low EQ (*Figure 4B*, *Supplementary file 2*). Concerning the reconstruction of side descent strategy based on hallucal proportions, our model reconstructed the omomyiform *Archicebus achilles* with a low side descent proportion (23%, *Figure 4D*, *Supplementary file 4*, *Supplementary file 5J*), despite its small size (~25 g) but relatively very long hallux (50% of foot length), similar to extant strepsirrhines. The same holds for the adapiforms *Darwinius masillae* and *Europolemur kelleri* that were estimated with moderate side descent proportions (33 and 55%, respectively, *Figure 4D*, *Supplementary file 5J*). Finally, the plesiadapiforms *Carpolestes simpsoni* and *Plesiadapis insignis*, with hallucal proportions similar to the studied platyrrhines, were reconstructed with relatively high side descent proportions (respectively, 58 and 75%, *Figure 4D*, *Supplementary file 5J*).

Finally, we investigated if the proportion of head-first descent on vertical supports follows a particular evolutionary pattern. Based on the literature, we included extinct taxa in the phylogeny of extant

species (*Figure 5*) and compared several quantitative models of evolution using the Akaike Information Criterion (AIC). We considered simple models commonly included in studies of morphological evolutionary dynamics: a single-rate Brownian Motion model (BM), the Early Burst model (EB), and a single-peak Ornstein-Uhlenbeck model (OU). We also considered models with two optima: a multiple-rate BM model (BMM) and a multiple-peak OU model (OUM). We considered various relevant phylogenetic nodes where peak shifts or rate changes could have occurred. We found that OUM models with a peak shift for 'euprimates + plesiadapiforms' or 'euprimates only' yielded the best fits (AIC = 9.46 and 9.48, respectively) both clearly favored from the other evolutionary models tested (*Supplementary file 5K*). These models have a head-first descent proportion theta peak of ~44 and ~45%, respectively, for a theta peak of ~95 and~93% for other mammal groups, respectively. After rescaling the phylogenetic tree height to 1, the strength of return to the optimum is quite high (alpha = 7.49 and 7.51 for the 'euprimates +plesiadapiforms' and 'euprimates only' models, respectively) and the phylogenetic half-life is small (0.093 and 0.092, respectively), confirming that these OUM models fit our dataset best (*Cooper et al., 2016*).

## Discussion

### The variability of vertical descent strategies in primates compared to other mammals

To date, this study is the first comparative quantification of ascent and descent strategies on vertical supports across a broad sample of small- to medium-sized arboreal mammals. Overall, our results reveal that vertical descent strategies are strongly influenced by phylogenetic relationships and not solely by body mass, contrary to our initial hypothesis (H1). Specifically, not all species >1 kg systematically descended tail-first. Primates exhibited greater variability in descent strategies—including head-first, side, and tail-first postures—whereas scandentians, rodents, carnivorans, and marsupials primarily employed head-first descents, with the exception of raccoons. Both head-first and tail-first postures involve the body remaining parallel to the support, as previously documented in rodents (*Karantanis et al., 2018*; *Youlatos, 2011*; *Youlatos and Panyutina, 2014*), procyonids (*McClearn, 1992*; *Jenkins and McClearn, 1984*), prehensile-tailed platyrrhines (*Youlatos and Gase, 1994*; *Garber and Rehg, 1999*; *Lawler and Stamps, 2002*), and non-prehensile-tailed strepsirrhines and marsupials (*Perchalski, 2021*; *Gaschk et al., 2019*). However, we also identified a distinct 'side descent' posture, observed exclusively in primates, in which the body is rotated perpendicular to the support while the face remains directed downward. This posture likely partly corresponds to the 'other descent strategies' described by *Perchalski, 2021*. We quantified side descents in both strepsirrhines and platyrrhines, with non-callitrichid platyrrhines using it most frequently (>70%) across all support diameters. In contrast, the smallest primates (i.e. pygmy slow loris *N. pygmaeus*, ~360 g; and tamarins *S. imperator* ~475 g, *S. oedipus* ~410 g) primarily employed head-first descents, while larger strepsirrhines, particularly lemurs, used tail-first descents in >80% of trials across all support diameters. Primates generally increased the use of tail-first and side descents on small vertical supports, suggesting that narrow supports impose additional constraints favoring more upright postures. This marked behavioral divergence among phylogenetic groups underscores the role of evolutionary history and supports recent findings emphasizing the influence of shared ecological niches and postural adaptations in shaping primate locomotor performance (*Granatosky et al., 2019*; *Shapiro et al., 2025*; *Dunham et al., 2019*; *Janisch et al., 2024*; *Wimberly et al., 2021*).

### The kinematic adjustments of arboreal mammals during vertical descents

In terms of locomotion, we found that all species reduced their speed and increased both forelimb and hindlimb duty factors during head-first descents compared to ascents on vertical supports, partially supporting our second hypothesis (H2). This pattern, previously observed in small primates (*Shapiro et al., 2016*; *Hesse et al., 2015*; *Nyakatura et al., 2008*), rodents (*Karantanis et al., 2018*; *Karantanis et al., 2017*; *Wölfer et al., 2021*), and marsupials (*Shapiro et al., 2014*; *Lammers et al., 2006*; *Gaschk et al., 2019*), suggests that head-first descent imposes greater demands on maintaining secure grips, leading to more cautious movements regardless of support diameter, characterized by increased limb contact times and reduced velocity (*Lammers and Stakes, 2025*; *Wölfer et al., 2021*).

However, when employing side or tail-first descent strategies, primates did not significantly reduce absolute speed nor markedly increase duty factors, but instead decreased relative distance traveled per cycle. This unexpected result suggests that tail-first and side descent strategies are highly effective, allowing primates to maintain comparable speeds while adopting flexible manual and pedal postures (*Toussaint et al., 2020*; *Toussaint, 2018*). During vertical ascent, locomotion primarily relies on hindlimb propulsion (*Hirasaki et al., 2000*; *Hanna and Schmitt, 2011*), whereas head-first descent requires increased forelimb braking and frictional control at the extremities (*Hirasaki et al., 2000*; *Hesse et al., 2015*; *Nyakatura et al., 2008*; *Nyakatura and Heymann, 2010*; *Preuschoft, 2002*; *Preuschoft et al., 1995*). Our results suggest that the upright descent strategies employed by primates mitigate these braking demands on the forelimbs while preserving static stability and high descent velocity.

Moreover, we found that gait patterns are strongly constrained by phylogenetic history, consistent with previous findings in primates (*Granatosky et al., 2019*; *Shapiro et al., 2025*; *Dunham et al., 2019*; *Janisch et al., 2024*; *Wimberly et al., 2021*). Primates, marsupials, rodents, and carnivorans exhibited both symmetrical and asymmetrical gaits, though primates and marsupials employed the highest proportions of symmetrical gaits, particularly DSDC and LSDC gaits. This aligns with earlier reports of habitual DSDC gait use in primates, some marsupials, and carnivorans, specifically the kinkajou *Potos flavus* (*Schmitt and Lemelin, 2002*; *Hildebrand, 1967*; *Cartmill et al., 2007b*; *Cartmill et al., 2002*; *Gaschk et al., 2019*; *Karantanis et al., 2015*; *Granatosky et al., 2016*). Interestingly, only strepsirrhines increased their use of LSLC gaits during descents, as previously observed in platyrrhines on sloped supports (*Nyakatura et al., 2008*; *Granatosky et al., 2019*; *Nyakatura and Heymann, 2010*; *Prost and Sussman, 1969*; *Rollinson and Martin, 1981*; *Shapiro et al., 2025*), whereas rodents and carnivorans favored more DSDC gaits during descents compared to more LSDC gaits in ascents. Both DSDC and LSLC gaits have been proposed to enhance stability by facilitating precise fore- and hindfoot placement, especially during horizontal walking on narrow branches (*Schmitt and Lemelin, 2002*; *Cartmill et al., 2002*; *Shapiro and Raichlen, 2007*). On the other hand, LSDC gaits promote stability by keeping the location of the center of mass relative to the support polygon of the limbs (*Shapiro and Raichlen, 2005*; *Cartmill et al., 2007b*; *Cartmill et al., 2002*; *Rollinson and Martin, 1981*; *Shapiro and Raichlen, 2007*; *Lammers and Zurcher, 2011*). Most previous studies have focused on symmetrical gaits, particularly DS, due to their proposed adaptive significance for fine-branch locomotion in primates (*Hildebrand, 1967*; *Cartmill et al., 2002*; *Granatosky et al., 2019*). However, several studies have also reported increased use of asymmetrical and LS gaits on sloped and vertical supports in primates and other arboreal mammals, emphasizing the importance of behavioral flexibility rather than strict kinematic specialization in arboreal environments to cope with complex 3D environments (*Karantanis et al., 2018*; *Wölfer et al., 2021*; *Granatosky et al., 2019*; *Nyakatura and Heymann, 2010*; *Granatosky et al., 2016*; *Granatosky, 2018*).

We found that the proportion of symmetrical gaits increased on small vertical supports, partially validating our second hypothesis (H2), but only during ascents. During descents, the proportion of asymmetrical gaits significantly increased across all supports and species, regardless of descent strategy. This key finding suggests that vertical locomotion requires distinct kinematic adjustments: symmetrical gaits enhance climbing efficiency, whereas asymmetrical gaits facilitate stability during descents. Symmetrical gaits, characterized by regular stance and swing phases, allow for controlled force application, lower reaction forces, control and transfer of the moments and torques imposed on the body axes, and minimized mediolateral deviations, keeping the center of mass within the support polygon throughout the stride (*Hildebrand, 1967*; *Cartmill et al., 2007b*; *Cartmill et al., 2002*; *Lemelin and Cartmill, 2010*). Conversely, asymmetrical gaits permit higher speeds, reduce metabolic cost, provide an energetic input to locomotion through the sagittal movement of the spine, and stabilize locomotion by reducing peak forces and center of mass oscillations through shorter, more frequent strides (*Shapiro et al., 2016*; *Karantanis et al., 2017*; *Wölfer et al., 2021*; *Hildebrand, 1977*; *Dunham et al., 2020*; *Chadwell and Young, 2015*). Overall, while symmetrical gaits enhance stability on narrow supports during horizontal or oblique locomotion and during vertical climbing (*Hanna et al., 2017*; *Karantanis et al., 2017*; *Lemelin and Cartmill, 2010*; *Nyakatura and Heymann, 2010*; *Cartmill et al., 2020*; *Karantanis et al., 2015*), asymmetrical gaits may offer superior control and security during vertical descents by enabling simultaneous grasping with both fore- and hindlimbs

(*Nyakatura et al., 2008*; *Karantanis et al., 2017*; *Wölfer et al., 2021*; *Hildebrand, 1977*; *Dunham et al., 2020*; *Chadwell and Young, 2015*; *Granatosky et al., 2022*).

## The correlation between morphological adaptations and vertical descent strategies

We identified a clear influence of body proportions on head-first descent behavior in our dataset, with body mass, relative forelimb, hindlimb, and tail lengths, and relative head size all significantly affecting descent strategies. Heavier animals with relatively larger heads as well as elongated limbs and tails were more likely to avoid head-first descents, partially supporting our third hypothesis (H3). As predicted, the intermembral index (IMI) was negatively correlated with the proportion of head-first descents (*Preuschoft, 2002*; *Perchalski, 2021*). This relationship was largely driven by strepsirrhines, which exhibited particularly low IMI values compared to other taxa. Strepsirrhines also performed the fewest head-first descents and the highest proportion of tail-first descents, highlighting the impact of hindlimb elongation on vertical locomotor performance by shifting the center of mass posteriorly. In contrast, platyrrhines displayed higher IMI values (>80), and all non-primate species exhibited relatively shorter limbs (forelimbs <55% and hindlimbs <70% of body length) compared to primates. These findings support previous hypotheses that elongated hindlimbs may increase the risk of imbalance during head-first descent, by simultaneously increasing the forelimbs' braking demand and the hindlimbs' traction requirements (*Preuschoft, 2002*; *Perchalski, 2021*).

We also detected a significant relationship between tail length and descent strategy: primates generally possessed proportionally longer tails (>90% of body length) than the non-primate species. Although tails primarily function in balance during horizontal arboreal locomotion (*Young et al., 2015*; *Stevens, 2008*; *Larson and Stern, 2006*) and serve to stabilize the body during leaping or aerial maneuvers (*Fukushima et al., 2021*), increased tail length may also contribute to a posterior shift of the center of mass that discourages head-first descent.

Interestingly, we also found a negative correlation between relative head size (approximated by the EQ) and the proportion of head-first descents, suggesting that larger head mass may further bias animals toward more upright postures during vertical descent. To our knowledge, this study is the first to explicitly assess the influence of body mass distribution—and particularly relative head mass—in the context of arboreal locomotion. Within our dataset, primates, especially platyrrhines, exhibited higher encephalization quotients (EQ >1.5) and engaged in head-first descent postures less frequently than other mammals, which possessed lower EQ values (<1.2) and relied more on head-first strategies. A larger brain not only demands increased cerebral blood flow, which may impose physiological limitations during acrobatic arboreal behaviors, but also increases head mass, thereby raising the moment of inertia and shifting the center of mass anteriorly. These factors support our hypothesis that an enlarged head may mechanically constrain animals to avoid head-first vertical descent by increasing the risk of forward toppling during head-first postures, particularly in primates. However, it is noteworthy that certain larger arboreal primates (>5 kg), such as howler monkeys (*Youlatos and Gase, 1994*) and orangutans (*Thorpe and Crompton, 2006*), both characterized by higher encephalization and, in the case of orangutans, markedly elongated forelimbs and elevated intermembral indices (*Fleagle, 2013*), have been observed performing head-first descents in their natural environments. Although these behaviors appear relatively infrequent, they underscore the need for further investigation into how brain expansion and associated morphological changes interact to influence arboreal locomotor strategies across taxa.

Contrary to our expectations (H3), we did not find a significant correlation between the relative size of the autopods and the proportion of head-first descents. However, within primates, we found a negative correlation between the hallux length and the proportion of side descents, with the longer the hallux, the less side descents employed. Digital elongation of the hand and foot in primates has long been interpreted as enhancing grasping efficiency on arboreal supports (*Nyakatura, 2019*; *Toussaint et al., 2020*; *Cartmill, 1992*; *Kirk et al., 2008*; *Lemelin and Jungers, 2007*; *Young and Chadwell, 2020*). During vertical ascent, the superior point of attachment (i.e. the hands) ensures a critical hold, whereas in head-first descents, both hands and/or feet may ensure holding. In contrast, tail-first or side descents may allow primates to rely more extensively on the foot for secure hold, supporting most of the weight. Strepsirrhines possess a highly specialized foot with a very divergent elongated hallux, while some platyrrhines possess a less elongated hallux but bear claw-like nails on

their distal phalanges which they use to cling to tree bark similarly to other mammals. These differences in hand and foot morphology among primates are reflected in their vertical descent behavior: strepsirrhines more frequently adopt tail-first descents, relying more heavily on the foot as a secure point of support, whereas platyrrhines more often employ side descents, using both hands and feet to grasp the support efficiently. Taken together, our results highlight that efficient vertical descent is a complex phenomenon governed by the interplay of kinematic adjustments, locomotor flexibility, body mass distribution, limb proportions, grasping ability, and shared evolutionary history among species. Primates exhibit particular adaptations for vertical descent, including upright postures and a remarkable capacity for locomotor flexibility, which may derive from their ancestral adaptation to navigating supports of varying size and orientation (*Granatosky et al., 2019*). In contrast, other arboreal mammals appear to have been more restricted to the use of less diversified supports during their evolutionary histories.

## Implications for the reconstruction of early primates locomotion

Our inference model reconstructed fossil species as employing relatively high proportions of head-first descent (>75%), except for the two medium-sized adapiforms *Darwinius masillae* and *Europolemur kelleri* (estimated at ~30% and 60%, respectively), which already displayed relatively long hindlimbs and tail. This validates our fourth hypothesis (H4) and suggests that postural adaptations for vertical descent in primates evolved progressively. Although some species already reached relatively high body mass, such as the adapiform *Notharctus tenebrosus* (~9 kg, *Harrington et al., 2016*), the plesiadapiform *Plesiadapis sp.* (~2 kg, *Orliac et al., 2014*), or the rodent *Ischyromys typus* (~1.3 kg, *Bertrand and Silcox, 2016*), they retained low EQs and moderate limb elongation. Interestingly, the plesiadapoids *Carpolestes simpsoni* and *Plesiadapis insignis*, whose phylogenetic positions within Euarchonta remain debated (*Upham et al., 2019*; *Janecka et al., 2007*; *Chester et al., 2017*; *Silcox and López-Torres, 2017*), have been reconstructed with a high proportion of side descent, associated with their slightly elongated hallux similar to extant platyrrhines. Plesiadapiforms are central to the debate on primate origins as the family Plesiadapidae is often regarded as a stem or sister group to euprimates (*Chester et al., 2017*; *Silcox and Gunnell, 2008*; *Silcox et al., 2017*). However, plesiadapiforms are mainly reconstructed as generalized quadrupedal walkers and climbers (*Bloch and Boyer, 2002*; *Silcox and Gunnell, 2008*; *Boyer, 2009*; *Youlatos and Godinot, 2004*). In contrast, early primates (including both omomyiforms and adapiforms) are all reconstructed as vertical climbers and leapers at variable degrees (*Ni et al., 2013*; *Marigó et al., 2019*; *Franzen et al., 2009*). Nonetheless, our evolutionary analysis corroborates current phylogenetic hypotheses suggesting that euprimates and plesiadapiforms share an adaptive regime distinct from the other groups studied [best OUM model: euprimates +plesiadapiforms vs. other], highlighting that descent strategy on vertical supports was tightly constrained throughout evolution.

These results lead us to suggest a refinement of the existing scenario on the order of acquisition of early primate specializations. Considering that early euarchontoglires were probably small to very small (≤100 g *Silcox and López-Torres, 2017*) with shorter hindlimbs, autopods, and reduced brain size, it is plausible that they primarily used mostly head-first descents and asymmetrical gaits on vertical supports (*Karantanis et al., 2018*; *Wölfer et al., 2021*). As euprimates evolved enhanced grasping abilities, elongated hindlimbs and tail, and eventually larger brains, they likely began to adopt side and tail-first vertical descent postures. The evolutionary success of early primates likely resided in their ability to colonize environments which are mechanically more challenging and difficult to access by other animals, such as horizontal terminal branches and small vertical supports. The frequent use of vertical supports, such as trunks, lianas, climbing plants, or even tall grass thickets, favored the early acquisition of a divergent hallux and pollex, improving grasping ability (*Gebo, 2011*; *Szalay and Dagosto, 1988*; *Toussaint et al., 2020*). These supports were abundant in the tropical forests of the northern hemisphere during the Paleogene, the likely origin place and time of primates (*Upham et al., 2019*; *Williams et al., 2010*; *Pozzi et al., 2014*). The use of these supports may have facilitated rapid access to the canopy, provided platforms for scanning for food or predators, efficient escape routes and rapid changes of height, and possibly privileged access to a greater diversity of food sources (*Del Rio et al., 2017*). Autopodial innovations were followed by hindlimb elongation, likely enhancing leaping abilities as inferred in various early euprimate fossils (*Boyer et al., 2017*; *Ravosa and Dagosto, 2007*). Altogether, these hindlimb specializations, followed by a

progressive increase of the body and tail lengths and brain size, probably altered body mass distribution. This may have further promoted side- and tail-first vertical descent strategies, allowing efficient vertical navigation while maintaining environmental awareness. Such adaptations may have thus favored primates to frequently adopt upright postures when negotiating vertical supports in an early stage of their evolutionary history, while increasing the use of symmetrical gaits to negotiate other types of supports, such as narrow horizontal branches. Building on previous hypotheses on autopodial grasping evolution of primates (*Cartmill, 1992*; *Toussaint et al., 2015*; *Sussman, 1991*; *Peckre et al., 2019*), upright orthograde postures would have also freed the hand from locomotion, while the foot remained specialized for efficient support holding. Functional differences between hand and foot have emerged early in primates (*Wood Jones, 1916*; *Preuschoft, 2002*; *Cartmill, 1972*; *Patel et al., 2015*), with the anchoring foot playing a fundamental role in body support, and the hand redirected toward other activities, such as food grasping and manipulation, substantiated by the great diversity of manual postures in extant species (*Toussaint et al., 2020*; *Pouydebat et al., 2006*; *Boyer et al., 2013*).

### Limitations of experimental conditions and future directions

Every study that includes experiments with animals housed in zoos displays some limitations. Some enclosures required enrichment with new supports, and in some cases, the animals did not readily use the support provided (see the Materials and methods section). It is, therefore, possible that this experimental bias may have had an impact on the proportion of descent behaviors and the kinematics quantified for some species. For instance, although raccoons and coatis have been observed to descend head-first on small and medium supports in the wild (personal observations), they did not use them in their enclosures. Thus, quantifying vertical descent strategies as well as the relative frequency of vertical support use in the wild would provide more conclusive evidence regarding the vertical descent repertoire in those species. Moreover, comparing kinematics of vertical ascents and descents in more species of larger body mass (>5 kg) with varying morphological proportions, such as apes and large carnivorans, would help to better understand the complex parameters at play regarding mechanical adaptation to arboreal locomotion in larger mammals. Such studies will also allow the development of reconstruction models better adapted to large fossils, particularly those primates with relatively lower EQ than extant species, thereby refining our understanding of the evolution of those adaptations within this group.

## Materials and methods
### Experimental design
#### Animals studied

We collected data on animals housed in zoological parks except for the forest dormice *Dryomys nitedula,* which were wild caught (*Supplementary file 1*). In zoos, individuals were video recorded directly in their enclosures. All the enclosures were large enough to allow unconstrained locomotion of the animals. We placed non-treated wooden branches of variable diameters and orientations in each enclosure before filming to ensure equal access to a variety of support types. All studied individuals were adults, in good shape, and did not display any stereotypical behavior before or during the experiments. The two individuals of *Dryomys nitedula* were caught in their natural habitat in Vasilika, Greece, with the use of specialized wooden nest boxes positioned in trees (permit by the Hellenic Ministry of Environment and Energy 8817/242). They were immediately transferred and housed in glass terrariums (L=100; W=40; H=50 cm) enriched with tree branches at room temperature and with normal photoperiod (complying with the regulations of the Ethics Committee of the School of Biology, Aristotle University of Thessaloniki, Greece). After observations and filming directly in these terrariums, the animals were released at the same location they were originally captured. Animal observation was performed in compliance with the International Primatological Society (IPS) Ethical Guidelines for the Use of Nonhuman Primates in Research and the Association for the Study of Animal Behavior (ASAB) and the Animal Behavior Society (ABS) ethical guidelines for the use of animals in research (*Animal Behaviour, 2020*).

## Behavioral data collection

We focused on vertical supports (90°±5°). We categorized support diameters into three categories: small (support circumference < hand and foot lengths), medium (support circumference ≈ hand and foot lengths), and large (support circumference > hand and foot lengths) following previously described methods (*Toussaint et al., 2020*), resulting in three support types tested for each individual. Sequences were recorded using a portable video camera (Panasonic HC-V770 camcorder 120fps, full HD 1080 p). To ensure both a complete view and close-ups of the animals during locomotion, we also used three small action video cameras (Mobius ActionCam 60fps, HD 720 p) installed in three different view angles (frontal, lateral and ventral) and in relatively close distance (between 20 and 50 cm depending on the size of the animals) from each support type. After a habituation period, the individuals were recorded using an alternation of scan sampling and focal-animal sampling methods (*Altmann, 1974*) in continuous recording sessions of 10–30 min. Animals were filmed as they moved freely over supports, and we stimulated them by providing small pieces of food placed randomly along the supports when necessary. For nocturnal species (*Supplementary file 1*), observations were conducted either in artificial or real nocturnal conditions with the addition of red lighting to enable recording without disturbing the animals.

## Video analysis

We analyzed videos using Adobe Premiere Elements 12. All video analysis was performed by the same person (SLDT) to avoid inter-observer biases. We focused on locomotion only and excluded other behaviors. We analyzed a minimum of three and up to ten passages in ascent and ten passages in descent for each individual on each support type (i.e. small, medium, and large), resulting in up to thirty ascents and thirty descents analyzed for each individual, with the following exceptions: we could not obtain data for *Tupaia belangeri* on small supports, for *Graphiurus murinus* on small supports; for *Typhlomys chapensis* on medium supports, for *Procyon lotor* on small supports, and for *Nasua nasua* on medium and small supports. For *Tupaia belangeri*, *Graphiurus murinus*, and *Typhlomys chapensis*, these supports were not available during the data acquisition process. However, for *Procyon lotor* and *Nasua nasua*, these supports had been added and were available in the enclosures but were not used by the animals.

## Kinematic analysis of locomotion

To quantify the absolute speed of the individuals for each sequence of locomotion, we extracted two snapshots and their associated frame number from the beginning and the end of the movement, using videos with a lateral view relative to the support. We merged both images using the product layer functionality in Adobe Photoshop CS6. We then measured in ImageJ the distance travelled by the individuals during each stride using markers of known length positioned in each support as references. We calculated the associated speed as the ratio of this distance to the duration separating the snapshots.

We also calculated the relative speed for each sequence of locomotion as the ratio of the absolute speed to the body length of each animal studied, defined as the total length between the top of the nose to the base of the tail. We used snapshots of each individual to measure body lengths in ImageJ, using a scale marker as a metric reference positioned on top of the branches.

We analyzed the strides for each sequence of locomotion by recording the frame number of the first touchdown, the following lift-off and the subsequent touchdown of each limb on the support. We calculated the associated stride duration of each limb, using the known frame rate of the associated videos. We then calculated the duty factor (DF) of each limb as the percentage of time spent in contact with the support relative to the total stride duration. Forelimbs and hindlimbs duty factors were averaged to obtain the mean DF.

We classified the gait type following a previously described methodology (*Dunham et al., 2020*). We used the midpoints of each limb contact interval (i.e. the temporal midpoint between first touchdown and following lift-off). We categorized gaits as symmetrical when the temporal lag between left and right forelimbs and hindlimbs midpoints were both ≥40% and ≤60% of the total stride duration, and asymmetrical otherwise. Further categorization among symmetrical gaits were made using the limb phase, which corresponds to the percentage of total stride duration separating a hindlimb touchdown from its ipsilateral forelimb touchdown. We thus calculated the left and right limb phases

using the midpoints of forelimbs and hindlimbs contact intervals. Left and right limb phases were then averaged to obtain the mean limb phase for each stride. We categorized as lateral sequence – lateral couplet (LSLC) when the mean limb phase was ≥0% and <25%, as lateral sequence – diagonal couplet (LSDC) when ≥25% and <50%, as diagonal sequence – diagonal couplet (DSDC) when ≥50 and <75%, and as diagonal sequence – lateral couplet (DSLC) when ≥75% and ≤100%. Classification of asymmetrical gaits used the stance period, which corresponds to the percentage of limb pair total contact duration separating left and right limbs of the same pair. We classified as bounds the gaits when both forelimb and hindlimb stance periods were ≤10%, as half-bounds those when either (i) the forelimb stance period was ≥10% and the hindlimb stance period was ≤10%, or (ii) when the forelimb stance period was ≤10% and the hindlimb stance period was ≥10%, and as gallop when both forelimb and hindlimb stance period values were ≥10%.

## Morphological data collection

For each studied individual, we extracted three to five lateral view screenshots from the videos. On these images, we measured lengths in ImageJ, allowing us to calculate eight intrinsic body proportions (*Table 1*). We calculated the average proportions over screenshots for each individual, and ultimately the average proportions by species.

The mean body mass of each species studied was gathered from the literature (*Supplementary file 2*). We could have used the actual body masses of certain individuals studied in zoos, but we could not have access to this information for most species, so we decided to use the standard reference values from literature. Additionally, we calculated the mean encephalization quotient (EQ) of these species as a proxy for their relative head to body size ratio and to allow comparison with fossil specimens for which the EQ is commonly studied and provided in the literature (*Supplementary file 2*). Prior to the calculation of the EQ in extant species, we collected the mean endocranial volumes (ECV) either from the literature or from our own measurements in the case of species that were not previously documented (*Supplementary file 3*). For these species, we quantified the ECV from preserved skulls of specimens housed in the anatomical collections of the Museum für Naturkunde of Berlin, by filling their endocranial cavity with chia seeds of known volumic mass to infer their volume. We thus provide hereby the mean endocranial volumes for the treeshrew *Tupaia belangeri*, the rodents *Dryomys nitedula*, *Graphiurus murinus* and *Typhlomys cinereus*, and the marsupial *Petaurus breviceps* (*Supplementary file 3*).

For comparative purposes, we collected from the literature the reconstructed body mass and EQ of key fossil specimens of euarchontoglires with relatively complete postcranial and/or cranial remains (*Supplementary file 2*). Whenever possible, we also measured the same postcranial lengths as for extant species, using published photographs (see *Supplementary file 2* for associated references, and *Supplementary file 4* for body proportions calculated in fossils).

## Construction of a phylogenetic tree including fossil taxa

For illustration purposes in *Figure 1C*, we used a phylogeny of extant species from https://timetree.org/; *Kumar et al., 2017* which derives from an aggregation of molecular phylogenetic data.

For phylogenetic comparative analysis (phylogenetic signals and evolutionary model comparisons), we generated 1000 trees from the tip-date DNA-only version of mammalian trees of the studied extant species from https://vertlife.org/; *Upham et al., 2019*. We computed a consensus tree using the *consensus* function of the R package *ape* and made it strictly ultrametric by extending the tips with the *force.ultrametric* function from the R package *phytools*. To build and compare evolutionary models of the fossils' descent behaviors, we supplemented this tree of extant species with the extinct taxa studied (*Figure 5*) using estimated age and phylogenetic topology from literature (*Supplementary file 2*).

## Statistical analysis

Statistical analysis was performed with RStudio version 1.3.1093 (R version 3.6.3). R scripts used for computations and statistical analysis are available at https://github.com/SLDToussaint/Vertical_descents_mammals (copy archived at *Toussaint, 2026*).

## Proportions of descent strategies

To assess the effect of support diameter (e.g. small vs. medium vs. large) on the proportion of descent types between individuals of the same species, we performed either analysis of variance (ANOVA) or multivariate analysis of variance (MANOVA) tests (respectively, *aov* and *manova* functions in R) depending on the number of descent strategies exhibited (*Figure 1B*, *Supplementary file 5A*). To even out the testing power among species for which we had a different number of individuals, we bootstrapped 10 individuals and 10 descent events on each support category to perform the tests. P-values were then Benjamini-Hochberg corrected.

To assess the extent of phylogenetic information contained in the proportion of head-first descents on vertical supports, we computed Blomberg's K phylogenetic signal (*Blomberg et al., 2003*) using the *physignal* function from the *geomorph* package in R with 50000 iterations.

## Locomotion

To test for the global differences of speed, relative speed, and duty factor between ascents and each descent type, we performed two-sided Mann-Whitney U tests on 40 bootstrapped events (20 ascents and 20 descents) for all species on each support diameter. P-values were then Benjamini-Hochberg corrected. We also calculated the mean percentage of variation between ascents and each descent category (*Figure 2*, *Supplementary file 5B–D*). For tests over duty factor, we considered both fore-limbs and hindlimbs combined.

We calculated the frequency of all possible gait type values occurring for each individual on each support category, differentiating ascents and the different descent strategies when relevant. These frequencies were PCA-transformed to obtain linearly independent values which serve as low-dimensional representations of the overall gait data. Using the same *physignal* function as described above for phylogenetic signal computation of head-first proportion on vertical supports, we computed the $K_{mult}$ statistic (*Adams, 2014*) with 50,000 iterations on the PCA-transformed data of gaits, keeping only the first principal components that retain 95% of variance, to assess the extent of phylogenetic information contained in gaits.

To check for enough similarities between species of the same phylogenetic group as defined in *Figure 1C*, and dissimilarities between groups to justify pooling data at the group level, we performed MANOVAs with Benjamini-Hochberg corrected post hoc pairwise comparisons on PCA-transformed frequencies of gaits, keeping principal components explaining at least 95% of variance, using the *manova* function from the *stats* package in R and Wilks's test (*Supplementary file 5E*).

To test for changes in the fraction of symmetrical versus asymmetrical gaits between ascents and descents for each group of species and on each support diameter separately, we performed two-sided Wilcoxon ranked signed tests on bootstrapped samples. For each group and each support, we took 100 bootstrapped species and sampled 20 events (10 ascents and 10 descents) at random. The resulting p-values were Benjamini-Hochberg corrected (*Figure 3*, *Supplementary file 5F*).

## Morphological correlates

To control that brain allometry is homogeneous among all phylogenetic groups, to be able to compare EQ between species, we computed major axis orthogonal regressions, following the recommendation of *Martin and Barbour, 1989*, between the Log transformed brain and body masses, over all species and by phylogenetic group using the *sma* package in R (*Figure 4—figure supplement 1*).

To assess the level of correlation among the 10 morphological variables, we computed Spearman correlations on all pairs. P-values were then Benjamini-Hochberg corrected (*Supplementary file 5G*).

We computed Spearman correlations to quantify the relation between morphological variables and the proportions of head-first descent (for all species) or side descent (for primates) on vertical supports (all diameters combined) and the resulting p-values were Benjamini-Hochberg corrected (*Figure 4*, *Figure 4—figure supplement 2*, *Supplementary file 5H, I*).

## Estimation of extinct species vertical descent strategy

We isolated the six morphological variables for which we obtained a significant correlation with head-first descent proportions on vertical supports (body mass, EQ, relative forelimb length, relative hind-limb length, intermembral index, and relative tail length). We used the Schafer's Multiple Imputation

procedure from the R package *norm* to compute estimations of the fossils' proportions of head-first descent given their respective morphological features (*Figure 4A–C*, *Figure 4—figure supplement 2A–C*, *Supplementary file 5J*). We simulated 5000 runs of Markov Chains with 50 steps to infer the head-descent proportions of these 13 extinct species. We conducted the same procedure to then infer the proportion of side descent in euprimates and plesiadapiform fossils by restricting the set of extant species to primates and using the only morphological variable which correlated with side descents (relative hallux to foot length, *Figure 4D*, *Supplementary file 5J*).

## Evolutionary model comparison for head-first descent strategy

To evaluate if the proportion of head-first descent strategy on vertical supports follows a specific evolutionary model, we compared different quantitative models of trait evolution in all extant and extinct taxa studied. We first fitted three simple models: Early Burst (EB), Brownian Motion (BM), and Ornstein-Uhlenbeck (OU) using the *mvEB, mvBM, and mvOU* functions from the R package *mvMorph,* respectively (*Clavel et al., 2015*). Then, we considered models with two distinct optima within the phylogeny. We fitted the Brownian Motion and Ornstein-Uhlenbeck models with multiple optima (respectively, BMM and OUM) using the *mvBM* and *mvOU* functions. We tested a change of the evolutionary optimum at different phylogenetic nodes: (I) euprimates, (II) euprimates + plesiadapiforms, (III) euarchontans (euprimates +plesiadapiforms + scandentians), and (IV) euarchontoglires (euarchontans + rodents). We compared models using the Akaike Information Criterion (AIC) and estimated the phylogenetic half-life in the OUM models that best fitted head-first descent strategies in our sample, using the *halflife* function from *mvMorph*, after rescaling the phylogenetic tree height to 1 (*Figure 5*; *Supplementary file 5K*).

## Acknowledgements

We greatly thank the staff of the Zoological and Botanical Park of Mulhouse, the Zoological Park of Paris, and the Spaycific'Zoo (France) for access to their animals and for their assistance during material installation and data collection. We thank Dr. A Panyutina and Dr. A Kuznetsov (Lomonosov Moscow State University and Russian Academy of Sciences) and the Moscow Zoo staff (Russian Federation) for video recordings of *Graphiurus murinus*, *Typhlomys chapensis*, and *Tupaia belangeri*. We thank Dr. NE Karantanis and the staff of the Nowe Zoo Poznan (Poland) for the video recordings of *Xantho-nycticebus pygmaeus*. We thank the staff of the Laboratoire d'Ecologie Générale in Brunoy (France) for the video recording of *Caluromys philander*. We thank C Funk from the Museum für Naturkunde of Berlin for access to the collection of mammal specimens. We thank Dr. A Llamosi for his help with statistical analysis.

## Additional information

### Funding

| Funder | Grant reference number | Author |
|---|---|---|
| Fondation Fyssen | postdoctoral grant | Severine LD Toussaint |
| Alexander von Humboldt Stiftung | postdoctoral grant | Severine LD Toussaint |
| Deutsche Forschungsgemeinschaft | NY63 2/1 | John A Nyakatura |
| Fonds Erasme | International Mobility Funds | Dionisios Youlatos |

The funders had no role in study design, data collection and interpretation, or the decision to submit the work for publication.

### Author contributions

Severine LD Toussaint, Conceptualization, Data curation, Formal analysis, Supervision, Funding acquisition, Investigation, Visualization, Methodology, Writing – original draft, Project administration;

Dionisios Youlatos, Supervision, Funding acquisition, Validation, Methodology, Writing – review and editing; John A Nyakatura, Conceptualization, Supervision, Funding acquisition, Validation, Investigation, Writing – review and editing

#### Author ORCIDs
Severine LD Toussaint https://orcid.org/0000-0002-5911-1122

Reviewer #1 (Public review): https://doi.org/10.7554/eLife.108268.3.sa1
Reviewer #2 (Public review): https://doi.org/10.7554/eLife.108268.3.sa2
Author response https://doi.org/10.7554/eLife.108268.3.sa3

## Additional files

#### Supplementary files
Supplementary file 1. List of animals studied. Species with an* are nocturnal. PZP = Parc Zoologique de Paris, France. PZBM=Parc Zoologique et Botanique de Mulhouse, France.

Supplementary file 2. Mean body mass, endocranial volume (ECV), and encephalization quotient (EQ) used for each extant and extinct species studied and their associated references. See *Supplementary file 3* for the details of ECV measurement conducted in this study.

Supplementary file 3. Endocranial volume (ECV) measured on specimens of various extant species. Specimens are from the anatomical collections of the Museum für Naturkunde of Berlin, Germany. We used Chia seeds, with a calculated reference of 1g=1.22 ml. We calculated the encephalization quotient (EQ) using a reference brain volumic mass of 1.036 g.cm-3 (*Ebinger, 1974*) and the following formula (*Boddy et al., 2012*): EQ=(1.036×ECV)/(0.056×(BodyMass)^0.746)

Supplementary file 4. Calculated limbs proportions for the studied extinct species. See *Supplementary file 2* for the associated mean body masses, mean encephalization quotient (EQ), and references from where we extracted the photographs used to measure the postcranial lengths. NA = not applicable

Supplementary file 5. Complete results of statistical analysis.

MDAR checklist

Source data 1. Raw behavioral and kinematic data, including the absolute speed, touchdowns and lift-offs used to quantify the duty factors and gait types, and body lengths measurements used to quantify the relative speed, by individual.

#### Data availability
All data needed to evaluate the conclusions of this study are included in this paper and its supplementary materials: behavioral and kinematics data as well as morphological data are provided in *Figures 1–4*, *Figure 3—figure supplement 1*, *Figure 4—figure supplements 1 and 2*, *Supplementary files 2–4*, *Source data 1*. R scripts used for computation and statistical analysis are available at https://github.com/SLDToussaint/Vertical_descents_mammals (copy archived at *Toussaint, 2026*). Given that a portion of the raw footage was acquired by collaborators (who granted us the rights to use it for this study only), and given the disproportionate work that would be needed to extract and catalog the very numerous sequences used here from their original files (which contain other non-published elements which are under study) we cannot make the raw videos available publicly at the time of publication. Nevertheless, raw video data can be available for scientific purposes only, upon reasonable request by email to SLDT and DY. They will consult other contributors that have authority over their content, and decide accordingly what can be shared considering the required effort to prepare data, the interest of the requester, and the legal options adapted (e.g. Data Use Agreement) to ensure that unpublished parts are protected.

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
