## [Editor Report · eLife Assessment]

This **valuable** study examines how mammals descend effectively and securely along vertical substrates. The conclusions from comparative analyses based on behavioral data and morphological measurements collected from 21 species across a wide range of taxa are **convincing**, making the work of interest to all biologists studying animal locomotion.

---

## [Referee Report · Reviewer #1 (Public review)]

Summary:

This unique study reports original and extensive behavioral data collected by the authors on 21 living mammal taxa in zoo conditions (primates, tree shrew, rodents, carnivorans, and marsupials) on how descent along a vertical substrate can be done effectively and securely using gait variables. Ten morphological variables reflecting head size and limb proportions are examined in relationship to vertical descent strategies and then applied to reconstruct modes of vertical descent in fossil mammals.

Strengths:

This is a broad and data-rich comparative study, which requires a good understanding of the mammal groups being compared and how they are interrelated, the kinematic variables that underlie the locomotion used by the animals during vertical descent, and the morphological variables that are associated with vertical descent styles. Thankfully, the study presents data in a cogent way with clear hypotheses at the beginning, followed by results and a discussion that addresses each of those hypotheses using the relevant behavioral and morphological variables, always keeping in mind the relationships of the mammal groups under investigation. As pointed out in the study, there is a clear phylogenetic signal associated with vertical descent style. Strepsirrhine primates much prefer descending tail first, platyrrhine primates descend sideways when given a choice, whereas all other mammals (with the exception of the raccoon) descend head first. Not surprisingly, all mammals descending a vertical substrate do so in a more deliberate way, by reducing speed, and by keeping the limbs in contact for a longer period (i.e., higher duty factors).

---

## [Referee Report · Reviewer #2 (Public review)]

Summary:

This paper contains kinematic analyses of a large comparative sample of small to medium-sized arboreal mammals (n = 21 species) traveling on near-vertical arboreal supports of varying diameter. This data is paired with morphological measures from the extant sample to reconstruct potential behaviors in a selection of fossil euarchontaglires. This research is valuable to anyone working in mammal locomotion and primate evolution.

Strengths:

The experimental data collection methods align with best research practices in this field and are presented with enough detail to allow for reproducibility of the study as well as comparison with similar datasets. The four predictions in the introduction are well aligned with the design of the study to allow for hypothesis testing. Behaviors are well described and documented, and Figure 1 does an excellent job in conveying the variety of locomotor behaviors observed in this sample. I think the authors took an interesting and unique angle by considering the influence of encephalization quotient on descent and the experience of forward pitch in animals with very large heads.

Comment from the Reviewing Editor on the revised version:

The authors responded to many comments of the reviewers, and I would be happy to see the authors make this version the Version of Record.

---

## [Author Response]

The following is the authors’ response to the original reviews.

**eLife Assessment:**
This valuable study examines how mammals descend effectively and securely along vertical substrates. The conclusions from comparative analyses based on behavioral data and morphological measurements collected from 21 species across a wide range of taxa are convincing, making the work of interest to all biologists studying animal locomotion.

We would like to greatly thank the two reviewers for their time in reviewing this work, and for their valuable comments and suggestions that will help to improve this manuscript.

Overall, we agree with the weaknesses raised, which are mainly areas for consideration in future studies: to study more species, and in a natural habitat context.

We will nevertheless add a few modifications to improve the manuscript, notably by making certain figures more readable, and adding definitions and bibliography in the main text concerning gait characteristics.

We also provide brief comments on each point of weakness raised by the reviewers below, in blue.

**Reviewer #1 (Public review):**
Summary:This unique study reports original and extensive behavioral data collected by the authors on 21 living mammal taxa in zoo conditions (primates, tree shrew, rodents, carnivorans, and marsupials) on how descent along a vertical substrate can be done effectively and securely using gait variables. Ten morphological variables reflecting head size and limb proportions are examined in relationship to vertical descent strategies and then applied to reconstruct modes of vertical descent in fossil mammals.Strengths:This is a broad and data-rich comparative study, which requires a good understanding of the mammal groups being compared and how they are interrelated, the kinematic variables that underlie the locomotion used by the animals during vertical descent, and the morphological variables that are associated with vertical descent styles. Thankfully, the study presents data in a cogent way with clear hypotheses at the beginning, followed by results and a discussion that addresses each of those hypotheses using the relevant behavioral and morphological variables, always keeping in mind the relationships of the mammal groups under investigation. As pointed out in the study, there is a clear phylogenetic signal associated with vertical descent style. Strepsirrhine primates much prefer descending tail first, platyrrhine primates descend sideways when given a choice, whereas all other mammals (with the exception of the raccoon) descend head first. Not surprisingly, all mammals descending a vertical substrate do so in a more deliberate way, by reducing speed, and by keeping the limbs in contact for a longer period (i.e., higher duty factors).Weaknesses:The different gait patterns used by mammals during vertical descent are a bit more difficult to interpret. It is somewhat paradoxical that asymmetrical gaits such as bounds, half bounds, and gallops are more common during descent since they are associated with higher speeds and lower duty factors. Also, the arguments about the limb support polygons provided by DSDC vs. LSDC gaits apply for horizontal substrates, but perhaps not as much for vertical substrates.

We analyzed gait patterns using methods commonly found in the literature and discussed our results accordingly. However, the study of limbs support polygons was indeed developed specifically for studying locomotion on horizontal supports, and may not be applicable for studying vertical locomotion, which is in fact a type of locomotion shared by all arboreal species. In the future, it would be interesting to consider new methods for analyzing vertical gaits.

The importance of body mass cannot be overemphasized as it affects all aspects of an animal's biology. In this case, larger mammals with larger heads avoid descending head-first. Variation in trunk/tail and limb proportions also covaries with different vertical descent strategies. For example, a lower intermembral index is associated with tail-first descent. That said, the authors are quick to acknowledge that the five lemur species of their sample are driving this correlation. There is a wide range of intermembral indices among primates, and this simple measure of forelimb over hindlimb has vital functional implications for locomotion: primates with relatively long hindlimbs tend to emphasize leaping, primates with more even limb proportions are typically pronograde quadrupeds, and primates with relatively long forelimbs tend to emphasize suspensory locomotion and brachiation. Equally important is the fact that the intermembral index has been shown to increase with body mass in many primate families as a way to keep functional equivalence for (ascending) climbing behavior (see Jungers, 1985). Therefore, the manner in which a primate descends a vertical substrate may just be a by-product of limb proportions that evolved for different locomotor purposes. Clearly, more vertical descent data within a wider array of primate intermembral indices would clarify these relationships. Similarly, vertical descent data for other primate groups with longer tails, such as arboreal cercopithecoids, and particularly atelines with very long and prehensile tails, should provide more insights into the relationship between longer tail length and tail-first descent observed in the five lemurs. The relatively longer hallux of lemurs correlates with tail-first descent, whereas the more evenly grasping autopods of platyrrhines allow for all four limbs to be used for sideways descent. In that context, the pygmy loris offers a striking contrast. Here is a small primate equipped with four pincer-like, highly grasping autopods and a tail reduced to a short stub. Interestingly, this primate is unique within the sample in showing the strongest preference for head-first descent, just like other non-primate mammals. Again, a wider sample of primates should go a long way in clarifying the morphological and behavioral relationships reported in this study.

We agree with this statement. In the future, we plan to study other species, particularly large-bodied ones with varied intermembral indexes.

Reconstruction of the ancient lifestyles, including preferred locomotor behaviors, is a formidable task that requires careful documentation of strong form-function relationships from extant species that can be used as analogs to infer behavior in extinct species. The fossil record offers challenges of its own, as complete and undistorted skulls and postcranial skeletons are rare occurrences. When more complete remains are available, the entire evidence should be considered to reconstruct the adaptive profile of a fossil species rather than a single ("magic") trait.

We completely agree with this, and we would like to emphasize that our intention here was simply to conduct a modest inference test, the purpose of which is to provide food for thought for future studies, and whose results should be considered in light of a comprehensive evolutionary model.

**Reviewer #2 (Public review):**
Summary:This paper contains kinematic analyses of a large comparative sample of small to medium-sized arboreal mammals (n = 21 species) traveling on near-vertical arboreal supports of varying diameter. This data is paired with morphological measures from the extant sample to reconstruct potential behaviors in a selection of fossil euarchontaglires. This research is valuable to anyone working in mammal locomotion and primate evolution.Strengths:The experimental data collection methods align with best research practices in this field and are presented with enough detail to allow for reproducibility of the study as well as comparison with similar datasets. The four predictions in the introduction are well aligned with the design of the study to allow for hypothesis testing. Behaviors are well described and documented, and Figure 1 does an excellent job in conveying the variety of locomotor behaviors observed in this sample. I think the authors took an interesting and unique angle by considering the influence of encephalization quotient on descent and the experience of forward pitch in animals with very large heads.Weaknesses:The authors acknowledge the challenges that are inherent with working with captive animals in enclosures and how that might influence observed behaviors compared to these species' wild counterparts. The number of individuals per species in this sample is low; however, this is consistent with the majority of experimental papers in this area of research because of the difficulties in attaining larger sample sizes.

Yes, that is indeed the main cost/benefit trade-off with this type of study. Working with captive animals allows for large comparative studies, but there is a risk of variations in locomotor behavior among individuals in the natural environment, as well as few individuals per species in the dataset. That is why we plan and encourage colleagues to conduct studies in the natural environment to compare with these results. However, this type of study is very time-consuming and requires focusing on a single species at a time, which limits the comparative aspect.

Figure 2 is difficult to interpret because of the large amount of information it is trying to convey.

We agree that this figure is dense. One possible solution would be to combine species by phylogenetic groups to reduce the amount of information, as we did with Fig. 3 on the dataset relating to gaits. However, we believe that this would be unfortunate in the case of speed and duty factor because we would have to provide the complete figure in SI anyway, as the species-level information is valuable. We therefore prefer to keep this comprehensive figure here and we will enlarge the data points to improve their visibility, and provide the figure with a sufficiently high resolution to allow zooming in on the details.

**Reviewer #1 (Recommendations for the authors):**
As indicated in the first section above, this is a strong comparative study that addresses important questions, relative to the evolution of arboreal locomotion in primates and close mammal relatives. My recommendations should be taken in the context of improving a manuscript that is already generally acceptable.(1) The terms symmetrical and asymmetrical gaits should be briefly defined in the main text (not just in the Methods section) by citing work done by Hildebrand and other relevant studies. To that effect, the statement on lines 96-97 about the convergence of symmetrical gaits is unclear. What does "Symmetrical gaits have evolved convergently in rodents, scandentians, carnivorans, and marsupials" mean? Symmetrical gaits such as the walk, run, trot, etc., are pretty the norm in most mammals and were likely found in metatherians and basal eutherians. This needs clarification. On line 239, the term "ambling" is used in the context of related asymmetrical gaits. To be clear, the amble is a type of running gait involving no whole-body aerial phase and is therefore a symmetrical gait (see Schmitt et al., 2006).

We have added a definition of the terms symmetrical and asymmetrical gaits and added references in the introduction such as: “Symmetrical gaits are defined as locomotor patterns in which the footfalls of a girdle (a pair of fore- or hindlimbs) are evenly spaced in time, with the right and left limbs of a pair of limbs being approximately 50% out of phase with each other (Hildebrand, 1966, 1967). Symmetrical gaits can be further divided into two types: diagonal-sequence gaits, in which a hindlimb footfall is followed by that of the contralateral forelimb, and lateral-sequence gaits, in which a hindlimb footfall is followed by that of the ipsilateral forelimb (Hildebrand, 1967; Shapiro and Raichlen, 2005; Cartmill et al., 2007b). In contrast, asymmetrical gaits are characterized by unevenly spaced footfalls within a girdle, with the right and left limbs moving in near synchrony (Hildebrand, 1977).” Now found in lines 87-94.

We corrected the sentence such as “Symmetrical gaits are also common in rodents, scandentians, etc..” Now found in line 107.

Thank you for pointing this out. We indeed did not use the right term to mention related asymmetrical gaits with increased duty factors. We removed the term « ambling » and the associated reference here. Now found in line 256.

(2) Correlations are used in the paper to examine how brain mass scales with body mass. It is correct to assume that a correlation significantly different from 0 is indicative of allometry (in this case, positive). That said, lines are used in Figure S2 that go through the bivariate scatter plot. The vast majority of scaling studies rely on regression techniques to calculate and compare slopes, which are different statistically from correlations. In this case, a slope not significantly different from 1.0 would support the hypothesis of isometry based on geometric similarity (as brain mass and body mass are two volumes). The authors could refer to the work of Bob Martin and the 1985 edited book by Jungers and contributions therein. These studies should also be cited in the paper.

Thank you for recommending us this better suited method. We replaced the correlations with major axis orthogonal regressions, as recommended by Martin and Barbour 1989. We found a positive slope for all species significantly different from 1 (0.36), indicating a negative allometry (we realized we were mistaken about the allometry terminology, initially reporting a “positive allometry” instead of a positive correlation).

We corrected in the manuscript in the Results and Methods sections, and cited Martin and Barbour 1989 such as:

“To ensure that the EQs of the different species studied are comparable and meaningful, we tested the allometry between the brain and body masses in our dataset following [84] and found a significant and positive slope for all species (major axis orthogonal regression on log transformed values: slope = 0.36, r^2^ = 0.92, p = 5.0.10^-12^), indicating a negative allometry (r = 0.97, df = 19, p = 2.0.10^-13^), and similar allometric coefficients when restricting the analysis to phylogenetic groups (Fig. S2).” Now found in lines 289-298.

- “To control that brain allometry is homogeneous among all phylogenetic groups, to be able to compare EQ between species, we computed major axis orthogonal regressions, following the recommendation of Martin and Barbour [84], between the Log transformed brain and body masses, over all species and by phylogenetic group using the sma package in R (Fig. S2).” Now found in lines 336-338.

We also changed Figure S2 in Supplementary Information accordingly.

(3) Trunk length is used as the denominator for many of the indices used in the study. In this way, trunk length is considered to be a proxy for body size. There should be a demonstration that trunk length scales isometrically with body mass in all of the mammals compared. If not the case, some of the indices may not be directly comparable.

We did not use trunk length as a proxy for body mass, but to compute geometric body proportions in order to test whether intrinsic body proportions could be related to vertical descent behaviors, namely the length of the tail and of the fore- and hindlimbs relative to the animal. We chose those indices to quantify the capability of limbs to act as levers or counterweights to rotate the animals for this specific question of vertical descent behavior. We therefore do not think that body mass allometry with respect to trunk length is relevant to compare these indices across species here. Also, we don’t expect that trunk length (which is a single dimension) would scale isometrically with body mass, which scales more as a volume.

(4) Given the numerous comparisons done in this study, a Bonferroni correction method should be considered to mitigate type I error (accepting a false positive).

We had already corrected all our statistical tests using the Benjamini-Hochberg method to control for false positives; see the SuppTables Excel file for the complete results of the statistical analyses. We chose this method over the Bonferroni correction because the more modern and balanced Benjamini-Hochberg procedure is better suited for analyses involving a large number of hypotheses.

(5) The terms "arm" and "leg" used in the main text and Table 1 are anatomically incorrect. Instead, the terms "forelimb" and hindlimb" should be used as they include the length sum of the stylopod, zeugopod, and autopod.

Indeed, thank you for pointing that out. We have corrected this error within the manuscript as well as in the figures 4 and S3.

(6) On p. 14, the authors make the statement that the postcranial anatomy of Adapis and Notharctus remains undescribed. The authors should consult the work of Dagosto, Covert, Godinot and others.

We did not state that the postcranial remains of Adapis and Notharctus have not been described. However, we were unfortunately unable to find published illustrations of the known postcranial elements that could be reliably used in this study. To avoid any misunderstanding, we removed the sentence such as: “However, we could not find suitable illustrations of the known postcranial elements of these species in the literature that could be reliably incorporated into this study. Thus, we only included their reconstructed body mass and EQ,..”. Now found in lines 393-397.

**Reviewer #2 (Recommendations for the authors):**
(1) Line 65/69 - Perchalski et al. 2021 is a single-author publication, so no et al. or w/ colleagues.

Indeed. This has been corrected in the manuscript, now found in lines 65 and 70.

(2) Lines 96-98 - Is it appropriate to say that the use of symmetrical gaits are examples of convergent evolution? There's less burden of evidence to state that these are shared behaviors, rather than suggesting they independently evolved across all those groups.

We agree with this and corrected the sentence such as “Symmetrical gaits are also common in rodents, scandentians, etc..” Now found in line 107.

(3) Line 198 - I am confused by how to interpret (-16,36 %) compared to how other numbers are presented in the rest of the paragraph.

To avoid confusion, we rephrased this sentence such as: “In contrast, primates did not significantly reduce their speed compared to ascents when descending sideways or tail-first (Fig. 2A, SuppTables B).” Now found in lines 207-209.